# Exercise-induced α-ketoglutaric acid stimulates muscle hypertrophy and fat loss through OXGR1-dependent adrenal activation

Yexian Yuan[1,†], Pingwen Xu[2,†], Qingyan Jiang[1,†], Xingcai Cai[1], Tao Wang[1], Wentong Peng[1], Jiajie Sun[1], Canjun Zhu[1], Cha Zhang[1], Dong Yue[1], Zhihui He[1], Jinping Yang[1], Yuxian Zeng[1], Man Du[1], Fenglin Zhang[1], Lucas Ibrahimi[2], Sarah Schaul[2], Yuwei Jiang[3], Jiqiu Wang[4], Jia Sun[5], Qiaoping Wang[6], Liming Liu[7], Songbo Wang[1], Lina Wang[1], Xiaotong Zhu[1], Ping Gao[1], Qianyun Xi[1], Cong Yin[1], Fan Li[1], Guli Xu[1], Yongliang Zhang[1] & Gang Shu[1,*,‡] (iD)

## Abstract

Beneficial effects of resistance exercise on metabolic health and particularly muscle hypertrophy and fat loss are well established, but the underlying chemical and physiological mechanisms are not fully understood. Here, we identified a myometabolite-mediated metabolic pathway that is essential for the beneficial metabolic effects of resistance exercise in mice. We showed that substantial accumulation of the tricarboxylic acid cycle intermediate α-ketoglutaric acid (AKG) is a metabolic signature of resistance exercise performance. Interestingly, human plasma AKG level is also negatively correlated with BMI. Pharmacological elevation of circulating AKG induces muscle hypertrophy, brown adipose tissue (BAT) thermogenesis, and white adipose tissue (WAT) lipolysis *in vivo*. We further found that AKG stimulates the adrenal release of adrenaline through 2-oxoglutarate receptor 1 (OXGR1) expressed in adrenal glands. Finally, by using both loss-of-function and gain-of-function mouse models, we showed that OXGR1 is essential for AKG-mediated exercise-induced beneficial metabolic effects. These findings reveal an unappreciated mechanism for the salutary effects of resistance exercise, using AKG as a systemically derived molecule for adrenal stimulation of muscle hypertrophy and fat loss.

**Keywords** AKG; lipolysis; obesity; OXGR1; thermogenesis
**Subject Categories** Metabolism; Musculoskeletal System

The EMBO Journal (2020) 39: e103304

## Introduction

Obesity is recognized as a global epidemic, and there is an urgent need to control obesity and obesity-related metabolic diseases (Gungor, 2014). Among diverse promising strategies for preventing obesity, physical exercise is considered to be one of the most effective ways of controlling body weight. Numerous intervention studies have evaluated the role of exercise in the attainment and maintenance of healthy body weight, as well as additional beneficial effects on metabolic, respiratory, and cardiovascular function independent of weight loss (DiPietro & Stachenfeld, 2000; Strasser, 2013). However, exercise presents variations in duration and intensity that promote different mechanical and metabolic stimuli, which result in distinct beneficial effects on cardiovascular function, whole-body metabolism, and glucose homeostasis.

Among the various classifications of exercise, endurance (aerobic) and resistance (nonaerobic) exercise are highlighted. Specifically, endurance exercise is a low-intensity and long-duration format of training, while resistance exercise is characterized by a high intensity and short duration (Patel *et al*, 2017). Endurance exercise is widely considered to increase endurance and cardiac

1  Guangdong Laboratory of Lingnan Modern Agriculture, Guangdong Province Key Laboratory of Animal Nutritional Regulation and National Engineering Research Center for Breeding Swine Industry, College of Animal Science, South China Agricultural University, Guangzhou, China
2  Division of Endocrinology, Department of Medicine, The University of Illinois at Chicago, Chicago, IL, USA
3  Department of Physiology and Biophysics, The University of Illinois at Chicago, Chicago, IL, USA
4  Ruijin Hospital, Shanghai Jiao Tong University School of Medicine, Shanghai, China
5  Zhujiang Hospital, Southern Medical University, Guangzhou, China
6  School of Pharmaceutical Sciences (Shenzhen), Sun Yat-Sen University Guangzhou, Guangzhou, China
7  State Key Laboratory of Food Science and Technology and Key Laboratory of Industrial Biotechnology, Ministry of Education, Jiangnan University, Wuxi, China
   *Corresponding author. Tel: +86 20 85284901; Fax: +86 20 85284901; E-mail: shugang@scau.edu.cn
   †These authors contributed equally to this work
   ‡Lead author

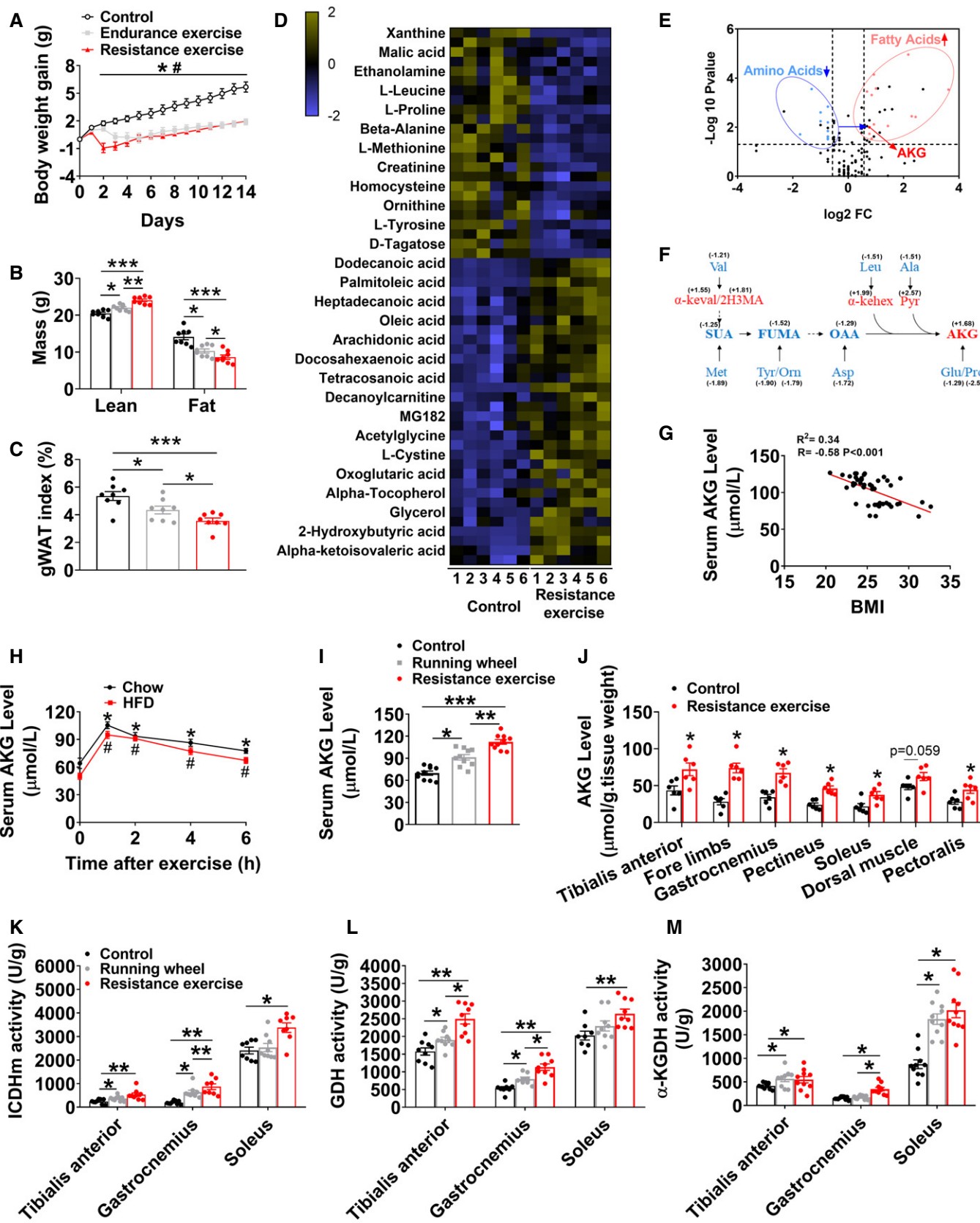

Figure 1.

**Figure 1.** AKG synthesis is induced by exercises.

A–C Mouse body weight gain (A), lean mass and fat mass (B), and gWAT index (C). At 8 weeks of age, male C57BL/6 mice were switched to HFD. After 12 weeks of HFD feeding, mice were divided into three groups receiving non-exercise, endurance exercise, or resistance exercise for 14 days (*n* = 8 per group).

D Relative changes in metabolites in response to resistance exercise. Heat maps show changes in metabolites in the serums from mice receiving resistance exercise or non-exercise. Male C57BL/6 mice (10 weeks old) fed with normal chow were divided into two groups receiving either non-exercise or resistance exercise for 40 min (*n* = 6 per group). Shades of yellow and blue represent fold increase and fold decrease in a metabolite, respectively (see color scale).

E A volcano plot of metabolome. Metabolites with $\log_2 FC \geq 0.58$ and $-\log_{10}P$ value $\geq 1.3$ were considered significant. Fatty acid (red dots) and amino acid (blue dots) metabolites were found to be significantly different between groups (*n* = 6 per group).

F Enrichment of tricarboxylic acid cycle (TCA cycle) intermediates/AKG metabolites in serum during resistance exercise. Blue color indicates significant decreases, while red color indicates significant increases by the volcano plot analysis between groups in serum metabolite levels (Val: valine; Leu: leucine; Ala: alanine; Met: methionine; Tyr: tyrosine; Orn: ornithine; Asp: aspartic acid; Glu: glutamic acid; Pro: proline; SUA: succinic acid; FUMA: fumaric acid; OAA: oxaloacetic acid; AKG: oxoglutaric acid; α-keval: alpha-ketoisovaleric acid; 2H3MA: 2-hydroxy-3-methylbutyric acid; α-kehex: α-ketoleucine; Pyr: pyruvic acid).

G Two-tailed Pearson's correlation coefficient analysis of plasma AKG level and body mass index (BMI) in Chinese adults (10 males and 35 females).

H Serum AKG concentration–time profile obtained before and after 40-min resistance exercise. At 8 weeks of age, male C57BL/6 mice were switched to HFD and continuously fed with HFD for 12 weeks. At 20 weeks of age, mice received resistance exercise for 40 min. Another group of chow-fed male C57BL/6 mice (10 weeks old) received resistance exercise for 40 min. The serum AKG level was tested at 0, 1, 2, 4, and 6 h after exercise (*n* = 8–10 per group).

I Serum AKG levels after exercise. Male C57BL/6 mice (10 weeks old) fed with normal chow were divided into three groups receiving non-exercise, free access to running wheel for 1 day, or resistance exercise for 40 min (*n* = 8–10 per group).

J Muscle AKG levels after exercise. Male C57BL/6 mice (10 weeks old) fed with normal chow were divided into two groups receiving either non-exercise or resistance exercise for 40 min (*n* = 6 per group).

K–M Muscle ICDHm (K), GDH (L), and α-KGDH (M) enzyme activity after exercise. Male C57BL/6 mice (10 weeks old) fed with normal chow were divided into three groups receiving non-exercise, free access to running wheel for 1 day, or resistance exercise for 40 min (*n* = 8–9 per group).

Data information: Results are presented as mean ± SEM. In (A), *$P \leq 0.05$ (Control versus Endurance exercise) and #$P \leq 0.05$ (Control versus Resistance exercise) by two-way ANOVA followed by post hoc Bonferroni tests. In (B–C and I–M), *$P \leq 0.05$, **$P \leq 0.01$, and ***$P \leq 0.001$ by one-way ANOVA followed by post hoc Tukey's tests. In (H), *$P \leq 0.05$ (Chow) and #$P \leq 0.05$ (HFD) by non-paired Student's *t*-test compared with before exercise.

health, while resistance exercise presents stimulatory effects on fat loss and muscle hypertrophy (Kilani, 2010). Although both endurance and resistance exercise lead to fat loss (Benito *et al*, 2015), resistance but not endurance exercise increases muscle mass and resting metabolic rate (Poehlman *et al*, 1991, 2002; Dolezal & Potteiger, 1998; Hunter *et al*, 2000), providing better weight loss maintenance in long-term observation. There have been numerous analyses of plasma metabolites following acute endurance exercise in both clinical and animal models (Lewis *et al*, 2010; Huffman *et al*, 2014; Aguer *et al*, 2017; Duft *et al*, 2017; Starnes *et al*, 2017; Sato *et al*, 2019), which has generated a number of metabolomics "signatures" in the circulation. These plasmas metabolic profiles provide signatures of endurance exercise performance and cardiovascular disease susceptibility and also identify molecular pathways that may modulate the salutary effects on cardiovascular function. However, very few metabolomics data are available for resistance exercise (Li *et al*, 2012; Berton *et al*, 2017), and the underlying chemical and physiological mechanisms for the stimulatory effects of resistance exercise on fat loss and muscle hypertrophy are not fully understood. Our goal is to identify the essential mediator for the beneficial metabolic effects of resistance exercise and provide potential therapeutic strategies to mimic the health effects of resistance exercise to combat obesity.

Emerging evidence has identified skeletal muscle as a secretory organ in regulating energy homeostasis and obesity progression in other tissues (Rai & Demontis, 2016; Ibrahim *et al*, 2017). Exercise can induce systemic metabolic effects either via changes in the mass and metabolic demand of muscle or via the release of muscle-derived cytokines (myokines) and metabolites (myometabolites) to target different downstream tissues (Schnyder & Handschin, 2015). Many myokines secreted in response to exercise improve glucose homeostasis and protect against obesity, such as irisin (Bostrom *et al*, 2012), interleukin-15 (IL-15) (Barra *et al*, 2014), and Meteorin-like (METRNL) (Rao *et al*, 2014). Similarly, myometabolites can mediate exercise-induced metabolic functions. For example,

β-aminoisobutyric acid (BAIBA), a novel exercise-induced muscle factor, attenuates insulin resistance, improves glucose tolerance, and promotes the browning of white adipose tissue (WAT) and hepatic β-oxidation (Roberts *et al*, 2014; Jung *et al*, 2015). In addition to their roles as metabolic substrates for gluconeogenesis, alanine and glutamine, the major amino acids released by skeletal muscle, can also act as hormone secretagogues and regulate the release of insulin, insulin-like growth factor 1, glucagon, and growth hormone (Nair & Short, 2005). Lactate, another prominent myometabolite released during exercise, has been proposed as a systemic modulator of metabolic homeostasis and the redox state (Brosnan & Letto, 1991; Corkey & Shirihai, 2012; Finsterer, 2012; Salgueiro *et al*, 2014). While metabolite therapies for obesity are emerging, metabolite-induced beneficial effects on improvement of obesity continue to face a serious challenge of low long-term therapeutic efficiency. Here, we aim to identify the essential exercise-induced myometabolites, which may mimic the long-term potent anti-obesity effects of regular physical exercise.

In the present study, we first applied a comparative metabolomics approach and demonstrated that substantial accumulation of a tricarboxylic acid (TCA) cycle intermediate, α-ketoglutaric acid (AKG), is a serum metabolic signature of acute resistance exercise in mice. We also found human plasma AKG is negatively correlated with BMI. We then systematically characterized the metabolic effects of AKG treatment in mice fed on chow or high-fat diet (HFD). Further, we used both loss-of-function and gain-of-function mouse models to determine whether AKG receptor OXGR1, expressed by the adrenal glands, is required for the anti-obesity effects of AKG. Finally, we tested whether OXGR1 is essential for resistance exercise-induced metabolic beneficial effects. Collectively, these results support the notion that AKG is an essential mediator of resistance exercise-induced beneficial metabolic effects. Notably, these data suggest that pharmacologically targeting the AKG–OXGR1 pathway may mimic some of the benefits of resistance exercise to improve metabolic health *in vivo*.

# Results

### Exercise induces the enrichment of AKG

Consistent with the previous observation that physical exercise can effectively decrease fat deposition (Maillard *et al*, 2018), we found that both ladder-climbing (resistance) and treadmill (endurance) exercise had similar effects of inhibiting HFD-induced body weight gain (Fig 1A). However, resistance exercise showed better beneficial metabolic effects compared with endurance exercise, as indicated by the higher lean mass and lower fat mass (Fig 1B) and gonadal adipose tissue (gWAT) index (Fig 1C). To search for the essential mediators of resistance exercise-induced metabolic salutary effects, we assessed the relative changes in serum metabolites in response to resistance exercise by using mass spectrometry to measure metabolites after resistance exercise in male mice. We obtained peripheral blood samples from both unexercised control mice and mice after acute resistance exercise (40 min at the conclusion of ladder-climbing exercise with 10% of body weight resistance). These training parameters had been shown to significantly increase serum lactate, a commonly used biomarker for peripheral muscle fatigue, indicating a successful resistance-training program (Fig EV1G). We found that 56 metabolites changed significantly at peak exercise compared to the unexercised group (Fig 1D). Most of the decreased metabolites are amino acids, while most of the increased metabolites are fatty acids (Fig 1D and E). Interestingly, several well-established accumulation signatures of succinate, malate, hypoxanthine, and xanthine induced by endurance exercise (Lewis *et al*, 2010) were found to be decreased by endurance exercise (Figs 1D and EV1A–D).

Additionally, the observed changes in plasma metabolites immediately after cessation of exercise reflect rapid upregulation of the TCA cycle intermediates/AKG-related metabolites (Fig 1F). Notably, AKG concentration in human plasma exhibited a statistically significant inverse relationship with several metabolic risk factors (Appendix Table S1), including body mass index (BMI, R = −0.58, $P < 0.001$; Fig 1G), hip circumference (HCF, R = −0.48, $P < 0.01$), waist circumference (WCF, R = −0.46, $P < 0.01$), fat mass (R = −0.42, $P < 0.01$), and body weight (R = −0.4, $P < 0.001$; Fig EV1H), suggesting an essential role of AKG in body weight control. We also showed that acute resistance exercise induced a time-dependent rapid increase in serum AKG in both chow- or HFD-fed mice (Fig 1H). Peak serum concentration was reached within 2 h after exercise and was 1.6 or 1.9 times higher than the physiological dose in non-exercise chow mice (105.41 ± 4.78 versus 64.11 ± 3.23 μM) or non-exercise HFD mice (94.93 ± 3.8 versus 50.13 ± 3.3 μM). Consistently, we found that a modest but significant increase in AKG was induced by wheel-running (endurance) exercise in chow-fed mice (91.327 ± 3.73 versus 69.801 ± 2.82 μM; Fig 1I). Importantly, resistance exercise induced a much higher increase in serum AKG level compared to wheel-running exercise (112.22 ± 3.16 versus 91.327 ± 3.73 μM; Fig 1I), suggesting an exercise type-dependent increase in serum AKG induced by exercise. Additionally, we found serum AKG level is not associated with running distance in wheel-running exercise (Fig EV1E), suggesting that exercise type instead of intensity plays a major role in the stimulation on serum AKG.

We further showed that AKG levels were consistently increased in different muscles from mice doing resistance exercise (Fig 1J). This result prompted us to examine whether exercise changes the activities of essential enzymes for AKG synthesis or degradation in the muscle. We tested several enzymes, including glutamate dehydrogenase (GDH), which converts glutamate to AKG; α-ketoglutaric acid dehydrogenase (α-KGDH), which catalyzes the conversion of AKG to succinyl-CoA; and isocitrate dehydrogenase (ICDHm), which catalyzes the oxidative decarboxylation of isocitrate, producing AKG and $CO_2$ (He *et al*, 2015; Xiao *et al*, 2016). We found both wheel-running and resistance exercise significantly enhanced the activities of all three enzymes in the tibialis anterior, gastrocnemius, and soleus (Fig 1K–M), suggesting that resistance exercise enhances AKG synthesis and release. This point of view is further supported by our observations that *in vivo* electrical stimulation of gastrocnemius muscle (hind limb) increased serum AKG (Fig EV1F). Thus, our observations indicate that exercise increases muscle AKG synthesis and blood AKG level, suggesting a physiological role of AKG in exercise-induced response.

### AKG mimics exercise-induced metabolic beneficial effects

If AKG plays a physiological role in exercise-induced beneficial effects, AKG supplementation will mimic some of the metabolic effects of exercise. Water supplementation of AKG is well tolerated

---

**Figure 2. AKG prevents diet-induced obesity.**

A Serum AKG concentration–time profile obtained from male C57BL/6 mice (10 weeks old) fed with normal chow before or after AKG gavage (10 mg/kg). The serum AKG level was tested at 0, 1, 2, 4, and 6 h after gavage (*n* = 8 per group).

B–D Water intake (B), cumulative food intake (C), and body weight gain (D) of male C57BL/6. At 12 weeks of age, mice were switched to HFD and received tap water or water supplemented with 2% AKG for 11 weeks (*n* = 9 per group).

E, F Fat and lean mass (E) and representative images (F) of body composition from male mice after 11 weeks of AKG supplementation (*n* = 9 per group).

G–I Weight index of BAT (G), gWAT (H), and iWAT (I) from male mice after 11 weeks of AKG supplementation (*n* = 9 per group).

J Serum AKG level of male mice after 11 weeks of AKG supplementation (*n* = 9 per group).

K, L Representative images (K) and quantification (L) of gWAT and iWAT HE staining from male mice after 11 weeks of AKG supplementation (*n* = 9 per group).

M–O Water intake (M), cumulative food intake (N), and body weight gain (O) of female C57BL/6 mice. At 12 weeks of age, mice were switched to HFD and received tap water or water supplemented with 2% AKG for 11 weeks (*n* = 9 per group).

P, Q Fat and lean mass index (P) and representative image (Q) of body composition from female mice after 11 weeks of AKG supplementation (*n* = 9 per group).

R–T Weight index of BAT (R), gWAT (S), and iWAT (T) from female mice after 11 weeks of AKG supplementation (*n* = 9 per group).

U, V Representative images (U) and quantification (V) of gWAT and iWAT HE staining from female mice after 11 weeks of AKG supplementation (*n* = 9 per group).

Data information: Results are presented as mean ± SEM. In (A), *$P ≤ 0.05$ by non-paired Student's *t*-test compared with before gavage. In (B–D and M–O), *$P ≤ 0.05$ by two-way ANOVA followed by post hoc Bonferroni tests. In (E, G–J, L, P, R–T and V), *$P ≤ 0.05$ and **$P ≤ 0.01$ by non-paired Student's *t*-test.

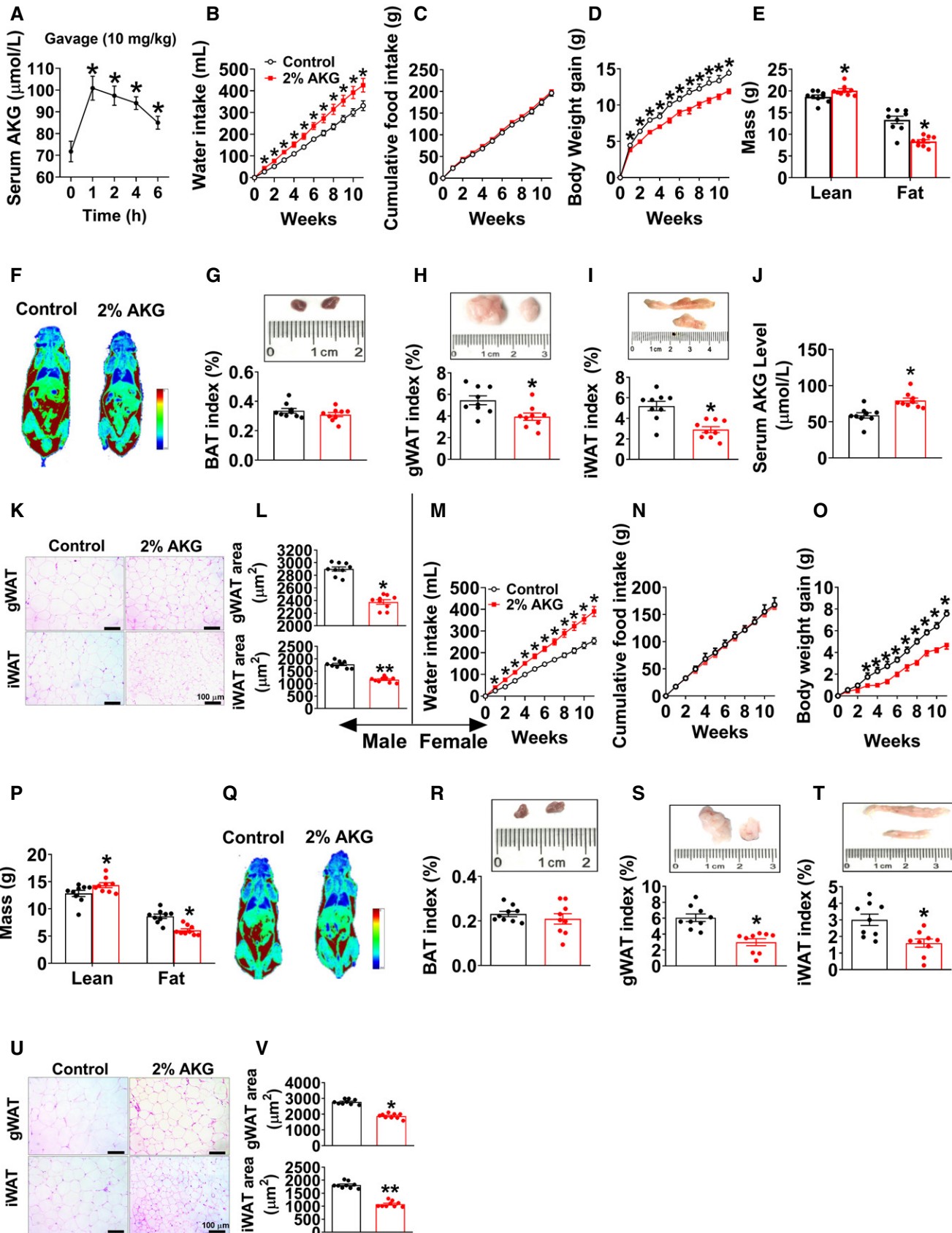

Figure 2.

(Chen *et al*, 2017). Moreover, we confirmed that acute oral administration resulted in increased circulating AKG (Fig 2A). On this basis, we systematically characterized the metabolic effects of 2% AKG supplementation in water in both male and female C57BL/6 mice fed on regular chow. We found that AKG significantly increased body weight gain in both male and female mice when fed chow (Fig EV2A and G). We also found increased food intake in both male and female AKG-treated mice (Fig EV2B and H). Notably, male or female AKG-treated mice started to gain more body weight than their controls at 1 or 2 weeks after treatment, whereas food intake differences began at 2 or 6 weeks after treatment (Fig EV2A, B, G and H), indicating that the hyperphagia phenotypes could be secondary to the body weight increase induced by AKG. These weight differences induced by AKG in both sexes were due to the increases in muscle size indicated by upregulated lean mass (Fig EV2C and I) and gastrocnemius and soleus weight (Fig EV2D and J). This is consistent with our previous observations that AKG promotes skeletal muscle hypertrophy and protein synthesis (Cai *et al*, 2016) while it inhibits skeletal muscle protein degradation and muscle atrophy (Cai *et al*, 2018).

Interestingly, opposite to the stimulatory effects on muscle mass, we found that AKG significantly decreased fat mass, weights of gWAT and inguinal white adipose tissue (iWAT), and adipocyte sizes of gWAT in both male and female mice (Fig EV2D–F and J–L). Consistent with decreased adiposity, we observed increased mRNA expression of thermogenic genes including uncoupling protein 1 (UCP1), iodothyronine deiodinase 2 (Dio2), and cell death-inducing DNA fragmentation factor-alpha-like effector A (Cidea) in brown adipose tissue (BAT) of AKG-treated male mice (Fig EV2M). Similarly, AKG-induced upregulation of UCP1 in the BAT was also suggested by both Western blot and immunohistochemistry (IHC) analyses of the UCP1 protein (Fig EV2N–P). These results suggest a role of AKG in BAT thermogenesis, which encourages us to examine if AKG regulates thermogenesis-related hormones. We found that AKG significantly increased serum epinephrine (E) and decreased non-esterified fatty acids (NEFA), but showed no effect on norepinephrine (NE), thyroxine (T4), or triiodothyronine (T3) in males (Fig EV2Q–U), implying an increased adrenergic stimulation induced by AKG. Thus, our observations indicate that AKG increases muscle mass and body weight, while at the same time decreasing WAT expansion and stimulating BAT thermogenesis of chow-fed mice.

To investigate whether AKG supplementation also produces beneficial metabolic effects in the diet-induced obesity (DIO) model, we characterized AKG's effects on energy homeostasis of male and female mice that were fed on HFD. Unexpectedly, in both sexes, AKG-treated mice showed increased water intake (Fig 2B and M), decreased body weight gain (Fig 2D and O), and comparable food intake (Fig 2C and N). Notably, both AKG-treated male and female mice still showed increased lean mass as we observed in chow-fed mice (Fig 2E, F, P and Q). The inhibitory effect of AKG on HFD-induced body weight gain was solely due to a decrease in fat mass, more specifically gWAT and iWAT but not BAT (Fig 2G–I and R–T). Consistently, in both sexes, the average adipocyte sizes of both gWAT and iWAT were significantly smaller in AKG-treated mice than in control mice (Fig 2K, L, U and V). Additionally, we found that 11 weeks of AKG water supplementation increased serum AKG level up to a dose comparable to that observed in HFD-fed mice receiving resistance exercise (increased from $58.77 \pm 3.2$ to $80.38 \pm 3.3$ μM, Fig 2J, versus $94.93 \pm 3.9$ μM, Fig 1H), suggesting a physiological boost of circulating AKG level. These data suggest a physiological role of AKG in preventing HFD-induced weight gain and expansion of adipose tissue in both sexes.

To determine the mechanisms underlying the protective effects of AKG on DIO, male control and AKG-treated mice were adapted into an indirect calorimetry system. AKG-treated mice showed significantly higher energy expenditure in both light and dark cycles (Fig 3A and B) compared to control mice. The enhanced energy expenditure was associated with increases in both core body temperature (Fig 3C) and cold-induced BAT temperature (Fig 3D and E), indicating an increase in thermogenesis. Consistently, AKG dramatically increased the mRNA expression of thermogenic genes, including UCP1, Dio2, and Cidea (Fig 3F), and the protein expression of UCP1 (Fig 3G–I) in the BAT. This AKG-induced BAT thermogenesis is further supported by the decreased serum NEFA in AKG-treated males (Fig 3J), suggesting a higher metabolism and oxidation rate of NEFA as an energy source. Combining our observation that AKG failed to affect cumulative HFD intake (Fig 2C) and calorie absorption (Fig EV3A), our data suggest that AKG prevents DIO by increasing thermogenesis and energy expenditure without affecting energy intake.

It has been shown that AKG lowered mouse body weight through influencing intestinal microbiota (Chen *et al*, 2017), suggesting another alternative mechanism for anti-obesity effects of AKG. On

---

**Figure 3. AKG increases fat thermogenesis and lipolysis.**

A, B  Oxygen consumption in male C57BL/6 mice after 11 weeks of AKG supplementation (*n* = 8 per group).
C  Body temperature of male mice after 11 weeks of AKG supplementation (*n* = 9 per group).
D, E  Representative images (D) and quantification (E) of BAT thermogenesis induced by 6-h cold exposure at 4°C in male mice supplemented with AKG for 11 weeks (*n* = 9 per group).
F, G  The mRNA expression of thermogenic genes (F) and immunoblots and quantification (G) of UCP1 protein in BAT of male mice after 11 weeks of AKG supplementation (*n* = 3–6 per group).
H, I  DAB staining (H) and quantification (I) of UCP1 in BAT of male mice supplemented with AKG for 11 weeks (*n* = 9 per group).
J  Serum levels of NEFA in male mice supplemented with AKG for 11 weeks (*n* = 9 per group).
K  The mRNA expression of CD137, CD40, TBX1, TMEM26, CITED1, and slc27a1 in iWAT of male mice supplemented with AKG for 11 weeks (*n* = 8 per group).
L, M  Respiratory exchange ratio (RER) in male C57BL/6 mice after 11 weeks of AKG supplementation (*n* = 8 per group).
N, O  Immunoblots (N) and quantification (O) of p-HSL and ATGL protein in gWAT of male mice after 11 weeks of AKG supplementation (*n* = 3 per group).
P, Q  Representative images (P) and quantification (Q) of p-HSL DAB staining in gWAT and iWAT of male mice after 11 weeks of AKG supplementation (*n* = 9 per group).
R  The mRNA expression of PPARγ, FASN, and ACC in the gWAT and iWAT from male mice supplemented with AKG for 11 weeks (*n* = 6 per group).
S–V  Serum levels of E (S), NE (T), T3 (U), and T4 (V) in male mice supplemented with AKG for 11 weeks (*n* = 8–9 per group).

Data information: Results are presented as mean ± SEM. In (B, C, E–G, I–K, M, O and Q–V), *P ≤ 0.05, **P ≤ 0.01, and ***P ≤ 0.001 by non-paired Student's *t*-test.

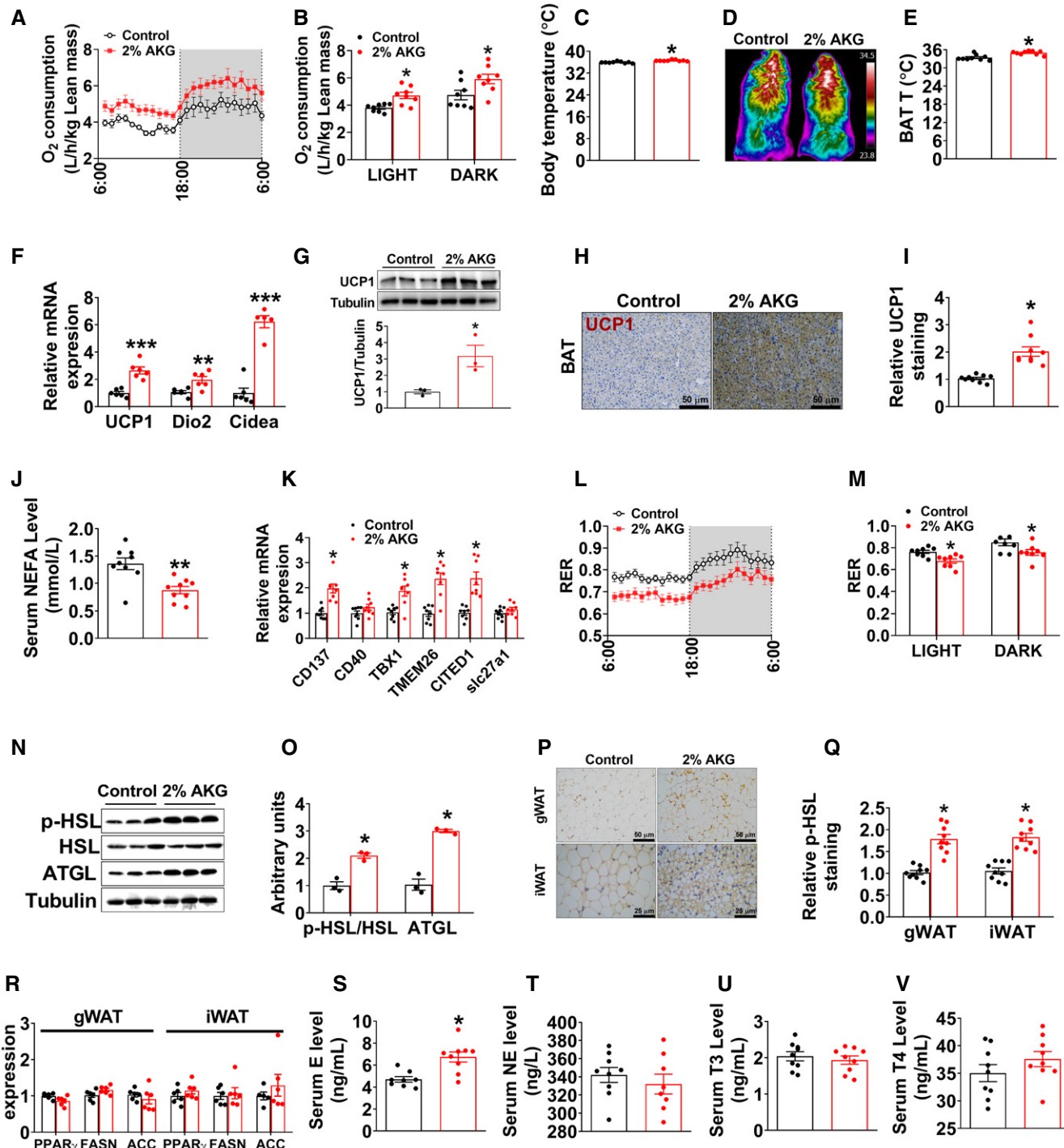

**Figure 3.**

this basis, we used 16S DNA sequencing to analyze the microbial composition in the feces from HFD-fed male mice after 1 or 4 weeks of AKG supplementation. Surprisingly, we found that AKG supplementation had no effect on microbial composition at both the phylum (Fig EV3B) and genus (Fig EV3C) levels analyzed by the ANOSIM and ADONIS methods (Fig EV3D). These suggest that microbial composition may not be the primary mediator for the inhibitory effects of AKG on HFD-induced obesity.

Notably, AKG also effectively increased the mRNA expression of brite (brown-in-white) markers in the iWAT, including tumor necrosis factor receptor superfamily member 9 (CD137), tumor necrosis factor receptor superfamily member 5 (CD40), T-box transcription factor 1 (TBX1), transmembrane protein 26 (TMEM26), Cbp/P300-interacting transactivator with Glu/Asp-rich carboxy-terminal domain 1 (CITED1), and solute carrier family 27 member 1 (slc27a1) in iWAT (Fig 3K). Additionally, we also found that when

maintained at room temperature (RT, 23°C), AKG-treated male mice formed brite adipocytes that expressed UCP1 protein (Fig EV3E–I). These results suggest that AKG supplementation enhances britening in the iWAT even at RT. To further assess the relationship between AKG- and cold-induced brite adipocyte formations, we exposed both control and AKG-treated male mice to a cold environment (6°C) for 1 week. Consistent with the well-established role of cold stimulus in brite formation (Berry *et al*, 2017), control mice exposed to cold showed increased UCP1-positive brite adipocytes (Fig EV3E–I). Interestingly, compared to cold-exposed control mice, AKG-treated mice exposed to cold showed significantly higher protein expression of UCP1 and smaller size of adipocytes (Fig EV3E–I), suggesting enhanced cold-induced brite formation. These results suggest that there may be distinct mechanisms for AKG- and cold-induced brite adipocyte formation.

Additionally, AKG also decreased the respiratory exchange ratio (RER) (Fig 3L and M), suggesting AKG-treated mice used more fat as a fuel source compared to control mice. Consistently, the increased RER was associated with enhanced lipolysis in the WATs, as indicated by increased phosphorylation of hormone-sensitive lipase (p-HSL) and protein expression of adipocyte triglyceride lipase (ATGL) in the WAT (Fig 3N–Q), both of which are the key lipases in adipocytes. Notably, a normal lipogenesis in WATs was indicated by unchanged mRNA expression of lipogenic genes (Fig 3R), including peroxisome proliferator-activated receptor gamma (PPARγ), fatty acid synthase (FASN), and acetyl-CoA carboxylase (ACC). Both BAT thermogenesis and WAT lipolysis are under coordinated control by metabolic hormones. Similar to what we observed in chow-fed mice, we found AKG significantly increased serum E, but not NE, T4, or T3, in males (Fig 3S–V), implying increased adrenergic stimulation induced by AKG. In summary, our results indicate that AKG stimulates BAT thermogenesis and fat metabolism and, by doing so, promotes energy expenditure and prevents DIO.

## Metabolic effects of AKG are mediated by adrenergic stimulation of adipose tissue thermogenesis and lipolysis

We next examined if acute AKG treatment would produce similar beneficial metabolic effects as we observed in the mice receiving long-term supplementation of AKG. Specifically, male C57BL/6 mice were intraperitoneally (i.p.) injected with AKG at a dose of 10 mg/kg. We found that this dose of AKG effectively increased blood AKG concentration up to a physiological level observed in resistance exercise mice within 2 h (Fig 4A). Interestingly, AKG administration also increased serum α-ketoleucine (α-kehex) and decreased fumaric acid (FUMA) (Fig EV4M–S), both of which are AKG derivatives and change after resistance training. Importantly, AKG acute treatment increased not only the temperature (Fig 4B and C) but also the expression of thermogenic genes, i.e., UCP1, Dio2, and Cidea (Fig EV4C and D), in the BAT 3 h after injection. Additionally, AKG also enhanced protein expression of ATGL and p-HSL in gWAT (Fig 4D) and decreased serum NEFA level (Fig EV4B). These results indicate that similar to long-term supplementation, acute i.p. injection of AKG also stimulates BAT thermogenesis and WAT lipolysis.

To explore the mechanism by which AKG promotes BAT thermogenesis, we examine the direct effects of AKG in *in vitro* or *ex vivo* models of BAT. We found that *in vitro* AKG treatment failed to affect mitochondrial function (Fig EV5A and B) and p-AMPKα or p-FoxO1 protein expression (Fig EV5I and J) of primary brown adipocyte. Consistently, we found that *ex vivo* AKG treatment failed to affect oxygen consumption rate (OCR) of BAT and NEFA levels in the culture medium (Fig EV5K and L), suggesting an indirect regulatory role of AKG in BAT metabolism. To further identify this indirect pathway, we evaluated the mitochondrial responses to AKG treatment in *in vitro* models of other metabolic organs, including chromaffin (adrenal gland), C2C12 (skeletal muscle), and HepG2 (liver) cell lines. We found that AKG decreased ATP production in all

---

**Figure 4.  Metabolic effects of AKG are mediated by adrenergic stimulation of thermogenesis and lipolysis.**

A       Serum AKG concentration–time profile obtained from male C57BL/6 mice (10 weeks old) fed with normal chow before or after i.p AKG (10 mg/kg body weight). The serum AKG level was tested at 0, 1, 2, 4, and 6 h after injection (n = 8 per group).

B, C   Representative images (B) and quantification (C) of BAT thermogenesis after 6-h cold exposure at 4°C. Male C57BL/6 mice (10 weeks old) were i.p. injected with 10 mg/kg AKG or saline and immediately exposed to cold stress at 4°C (n = 8 per group).

D       Immunoblots and quantification of p-HSL and ATGL in the gWAT of male C57BL/6 mice (10 weeks old) 3 h after i.p. injection of 10 mg/kg AKG or saline (n = 3 per group).

E       Serum E level in AKG-treated male mice 3 h after i.p. injection (n = 8 per group).

F–I    Oxygen consumption (F–G) and RER (H–I) in male C57BL/6 mice (10 weeks old) i.p. injected with saline, 10 mg/kg AKG, 1 mg/kg SR59230A (ADRB3 inhibitor), or AKG + SR59230A (n = 8 per group). All injections were performed at 7:00 am of the second day. Data are summarized in bar graph (G and I) by light or dark cycle of the second day.

J–N    Body weight gain (J), cumulative food intake (K), body composition (L), fat weight (M), and serum NEFA (N) of sham or adrenalectomized male C57BL/6 mice. Male mice were adrenalectomized at 8 weeks of age. Two weeks after surgeries, male mice were switched to HFD and given free access to tap water or 2% AKG for 9 weeks (n = 8 per group).

O, P   Representative images (O) and quantification (P) of BAT thermogenesis after 6-h cold exposure at 4°C in sham or adrenalectomized male mice treated with AKG for 9 weeks (n = 8 per group).

Q       The mRNA expression of thermogenic genes in the BAT of sham or adrenalectomized male mice treated with AKG for 9 weeks (n = 6 per group).

R, S   Immunoblots (R) and quantification (S) of p-HSL and ATGL protein in the gWAT of sham or adrenalectomized male mice treated with AKG for 9 weeks (n = 4 per group).

T, U   Immunoblots (T) and quantification (U) of UCP1 protein in the BAT of sham or adrenalectomized male mice treated with AKG for 9 weeks (n = 4 per group).

Data information: Results are presented as mean ± SEM. In (A), *$P \leq 0.05$ by non-paired Student's *t*-test compared with before injection. In (C–E), *$P \leq 0.05$ and **$P \leq 0.01$ by non-paired Student's *t*-test. In (J, K), *$P \leq 0.05$ by two-way ANOVA followed by post hoc Bonferroni tests. In (G, I, L–N, P, Q, S and U), different letters between bars indicate $P \leq 0.05$ by one-way ANOVA followed by post hoc Tukey's tests.

Source data are available online for this figure.

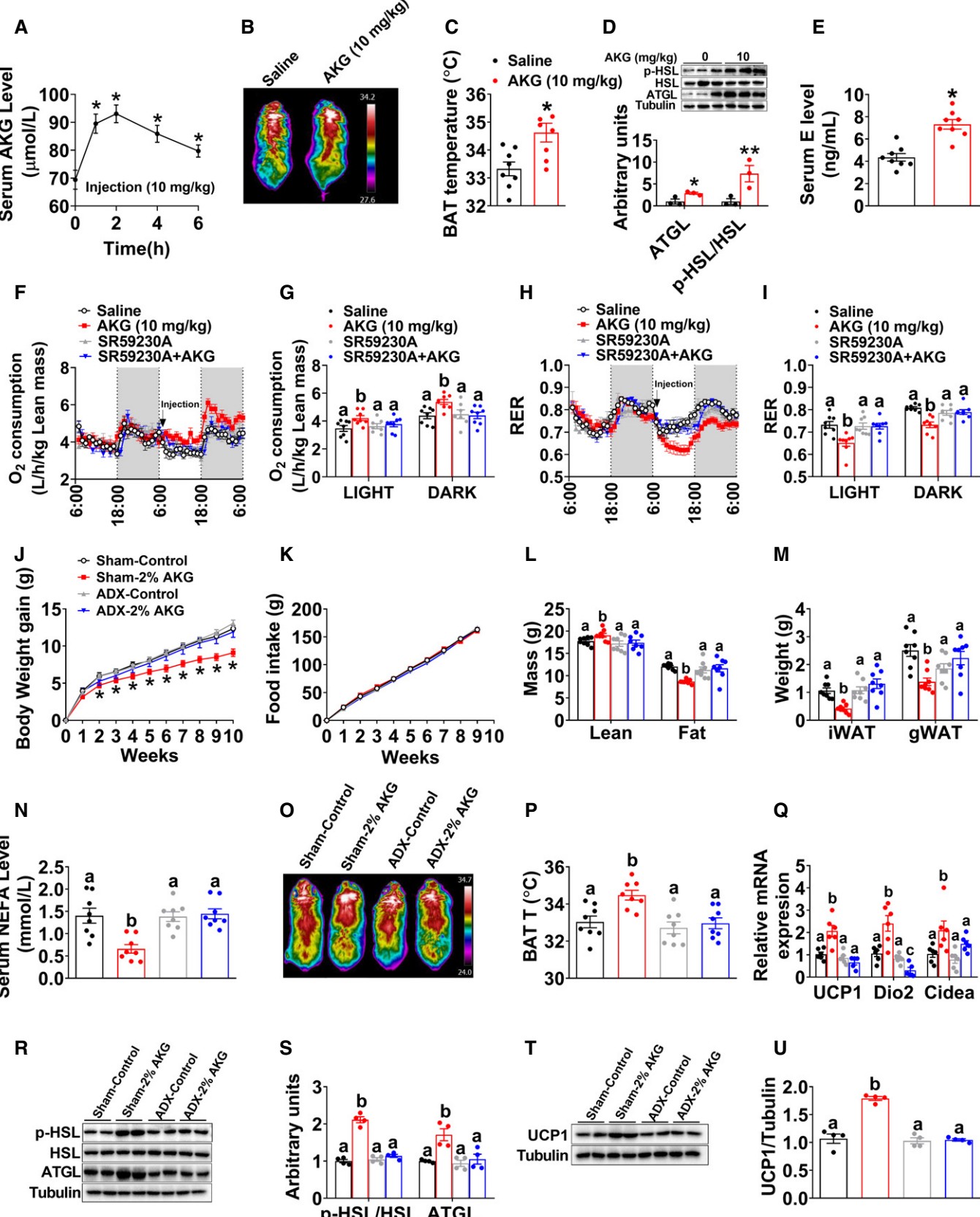

**Figure 4.**

models (Fig EV5C–H), which is consistent with a previous observation that AKG extends lifespan by inhibiting the ATP synthase (Chin *et al*, 2014). Additionally, AKG also decreased the basal respiration of C2C12 cells (Fig EV5E and F), suggesting an autocrine regulatory role of AKG in muscle metabolism, which is consistent with our previous findings (Cai *et al*, 2016, 2018). Importantly, AKG dramatically decreased basal respiration and enhanced spare respiratory capacity (SRC) of adrenal chromaffin cells (Fig EV5C and D), which enables cells to overcome various stresses including HFD-induced oxidation stress, suggesting a direct effect of AKG on the adrenal gland.

In supporting this view, AKG stimulated the release of instantaneous intracellular calcium from chromaffin cells (Fig EV5P), and this stimulatory effect is dose-dependent (Fig EV5Q), suggesting AKG-induced direct activation of intracellular calcium-dependent signaling pathways. Importantly, we also found AKG dose-dependently increased the release of E, but not NE, from chromaffin cells (Fig EV5R and S), indicating activation of adrenal medulla function. The same stimulatory effects were consistently observed in *ex vivo* adrenal gland model. Specifically, we found AKG increased the concentration of E, but not NE, in the medium from organ cultures of adrenal glands (Fig EV5M and N). Additionally, the protein expression of phospholipase C-β (PLCβ), one of the intracellular calcium signaling effectors, was enhanced in the adrenal gland by *ex vivo* AKG treatment (Fig EV5O). Therefore, our data suggest that AKG directly acts on adrenal medullary chromaffin cells to increase E release.

This view is further supported by the evidence from *in vivo* mouse model. Specifically, we found that protein expression of PLCβ and phosphorylation of extracellular-signal-regulated kinase (p-Erk) in the adrenal glands were upregulated by acute AKG treatment (Fig EV4E). The Erk pathway is involved in directing cellular responses to extracellular stimuli (Roberts, 2012). The upregulation of both PLCβ and p-Erk indicates enhanced adrenal activation. Notably, serum catecholamine (E but not NE) was significantly increased by both acute and long-term AKG treatments (Figs 3S and T, and 4E and EV4A), suggesting an AKG-

induced activation of the adrenal medulla. Consistently, AKG treatment also increased heart rate (Fig EV4H and I) and blood pressure (Fig EV4J–L), both of which are direct physiological and behavioral responses induced by adrenal gland E. However, no obvious difference was observed in locomotor activity (Fig EV4F and G). Taken together, both *in vitro* and *in vivo* evidence supports that AKG directly acts on the adrenal gland to increase the release of E.

It is well established that catecholamines are an essential driver of BAT thermogenesis by stimulating the UCP1 signaling pathway (Sharara-Chami *et al*, 2010). Importantly, catecholamines also induce WAT lipolysis to promote the release of fatty acids, which are used as the principal substrate for BAT thermogenesis (Bartelt *et al*, 2011). Therefore, increased serum E may mediate the stimulatory effects of AKG on BAT thermogenesis and WAT lipolysis. Consistent with this speculation, we found that while acute AKG treatment significantly increased oxygen consumption and decreased RER, both regulatory effects were abolished by co-injection of SR59230A, a beta-3 adrenergic receptor (ADRB3) inhibitor (Fig 4F–I). ADRB3 is the key mediator for the stimulatory effects of catecholamines on WAT lipolysis and BAT thermogenesis (Claustre *et al*, 2008; Jiang *et al*, 2017). These results suggest a mediating role of E in the anti-obesity effects of AKG.

To provide further evidence to support this hypothesis, we investigated the metabolic effects of AKG in adrenalectomized male mice. Interestingly, we found that the anti-obesity effects of AKG were abolished by adrenalectomy (Fig 4J–U). Specifically, not only the inhibition on body weight (Fig 4J), fat mass (Fig 4L), iWAT and gWAT weight (Fig 4M), and serum NEFA (Fig 4N), but also the stimulation on lean mass (Fig 4L), cold-induced BAT thermogenesis (Fig 4O and P), mRNA expression of thermogenic genes in the BAT (Fig 4Q), protein expression of ATGL and p-HSL in the iWAT (Fig 4R and S), and UCP1 protein expression in the BAT (Fig 4T and U) were diminished by adrenalectomy. These results suggest a mediating role of adrenal stimulation in AKG-induced adipose tissue lipolysis and thermogenesis.

---

**Figure 5. OXGR1 is required for the stimulatory effects of AKG on thermogenesis and lipolysis.**

A   The mRNA expression of AKG-sensing genes in the adrenal gland tissue of 12-week-old male C57BL/6 mice (*n* = 4 per group).
B   The mRNA expression of OXGR1 in different tissues of 12-week-old male C57BL/6 mice (*n* = 3 per group).
C   OXGR1 localization in adrenal gland medulla indicated by fluorescent staining of OXGR1 (green).
D   The mRNA expression of OXGR1 in the adrenal gland of male mice 3 h after i.p. injection of saline or 10 mg/kg AKG, or immediately after 40-min resistance exercise (*n* = 8 per group).
E   Immunoblots and quantification of OXGR1 protein expression in adrenal chromaffin cells treated with negative control (NC) siRNA or siOXGR1 (*n* = 3 per group).
F   E level in the medium from adrenal chromaffin cells cultured with vehicle + NC, vehicle + siOXGR1, AKG (100 μM) + NC or AKG + siOXGR1 for 30 min (*n* = 8 per group).
G   Intracellular calcium ion [Ca$^{2+}$] changes in adrenal medulla cells cultured with vehicle + NC, vehicle + siOXGR1, AKG (100 μM) + NC, or AKG + siOXGR1 (*n* = 30 per group).
H, I   Body weight gain (H) and cumulative food intake (I) of male WT control (littermates) or OXGR1 global knockout (OXGR1KO) mice. At 12 weeks of age, both control and KO mice were switched to HFD and further divided into two groups receiving tap water or water supplemented with 2% AKG for 13 weeks (*n* = 8 per group).
J, K   Representative images of body composition (J) and fat and lean mass index (K) of male WT or OXGR1KO mice treated with AKG for 13 weeks (*n* = 8 per group).
L, M   Weight index of iWAT (L) and gWAT (M) in male WT or OXGR1KO mice treated with AKG for 13 weeks (*n* = 8 per group).
N, O   Immunoblots (N) and quantification (O) of p-HSL and ATGL protein in gWAT of male WT or OXGR1KO mice treated with AKG for 13 weeks (*n* = 4 per group).
P–S   Oxygen consumption (P-Q) and RER (R-S) of male WT or OXGR1KO mice treated with AKG for 13 weeks (*n* = 8 per group).

Data information: Results are presented as mean ± SEM. In (A, B, F, K–M, O, Q, and S), different letters between bars indicate *P* ≤ 0.05 by one-way ANOVA followed by post hoc Tukey's tests. In (D), **\*\****P* ≤ 0.01 by one-way ANOVA followed by post hoc Dunnett's tests. In (E), **\*\****P* ≤ 0.01 by non-paired Student's *t*-test. In (G–I), \**P* ≤ 0.05 by two-way ANOVA followed by post hoc Bonferroni tests.
Source data are available online for this figure.

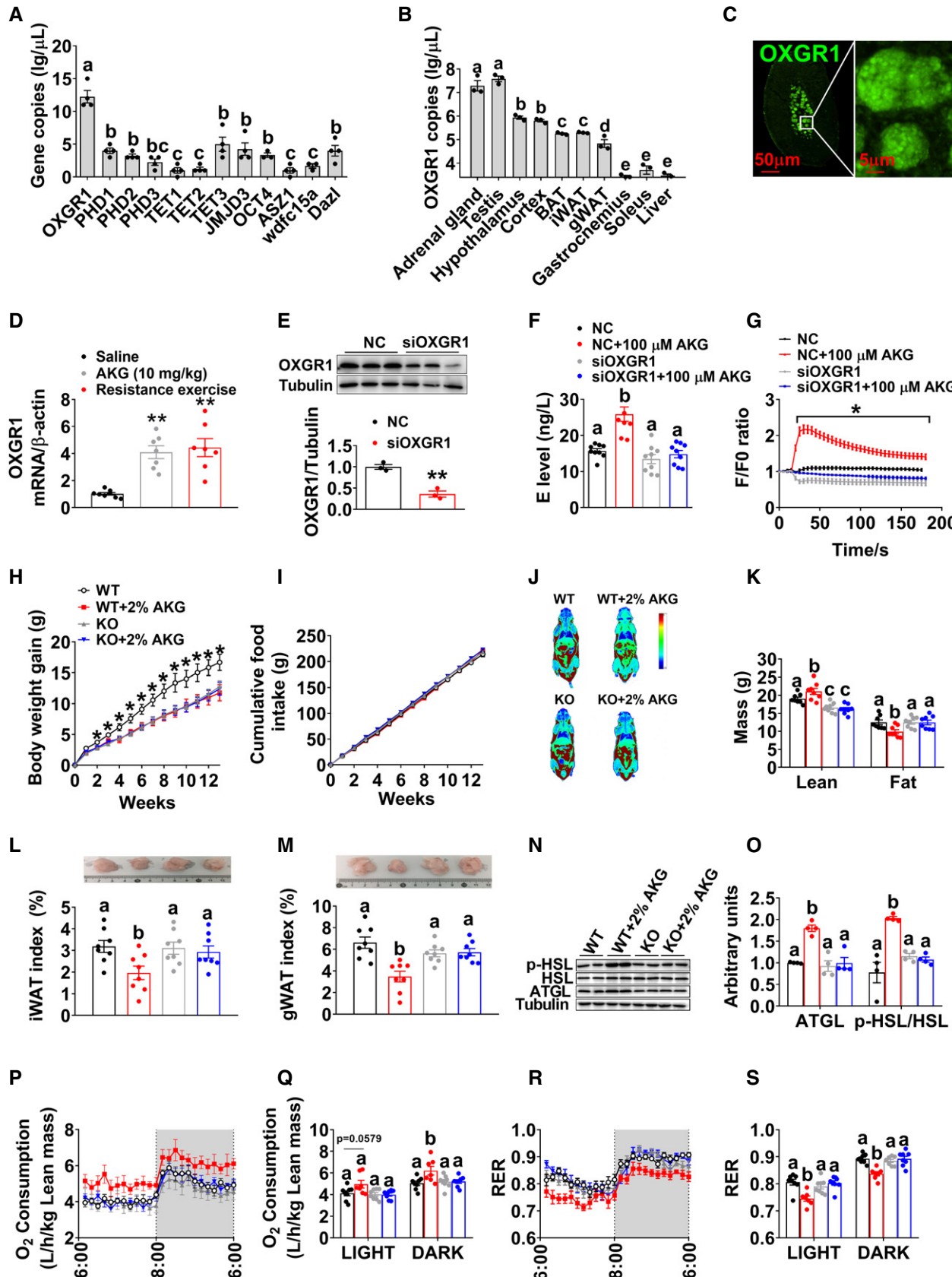

Figure 5.

## 2-Oxoglutarate receptor 1 in the adrenal gland (OXGR1^AG) is required for the stimulatory effects of AKG on adipose tissue thermogenesis and lipolysis

As an endogenous intermediate metabolite in the TCA cycle, AKG is traditionally known as an energy donor or a precursor in amino acid biosynthesis (Wu *et al*, 2016). However, recent studies have shown that AKG also functions as a signaling molecule and acts as a regulator of epigenetic processes and cellular signaling, via protein binding with many different AKG sensors (Zdzisinska *et al*, 2017). These AKG sensors include hypoxia-inducible factor prolyl-hydroxylases (PHDs), ten-eleven translocations (TETs), lysine demethylase 6B (JMJD3), octamer-binding transcription factor 4 (OCT4), ankyrin repeat, SAM and basic leucine zipper domain containing 1 (ASZ1), WAP four-disulfide core domain 15A (wdfc15a), depleted in azoospermia-like (Dazl), and its endogenous G protein-coupled receptor (OXGR1) (Zdzisinska *et al*, 2017). We postulate that the stimulatory effect of AKG on adrenal E secretion is mediated by one of these AKG sensors. Notably, we found that OXGR1 has the highest absolute mRNA expression level in the adrenal gland among different AKG sensors (Fig 5A). Adrenal OXGR1 absolute mRNA expression is also much higher than the expression in other tissues, except the testis (Fig 5B). Additionally, immunofluorescence staining (IF) of OXGR1 showed strong signals in the adrenal inner medulla instead of the outer cortex (Fig 5C). Interestingly, both acute AKG treatment and resistance exercise increased the mRNA expression of OXGR1 in the adrenal gland (Fig 5D). These results suggest a possible role of OXGR1^AG in the direct stimulatory effects of AKG on adrenal E release.

To test this point of view, we first generated a loss-of-function *in vitro* chromaffin cell model by using siRNA to target OXGR1 specifically. We found OXGR1 siRNA-treated chromaffin cells showed significantly less protein expression of OXGR1 compared to control siRNA-treated cells (Fig 5E), which validated our OXGR1-knockdown chromaffin cell model. By using this model, we showed that the knockdown of OXGR1 abolished the stimulatory effects of AKG on the secretion of E (Fig 5F) and release of instantaneous intracellular calcium (Fig 5G), suggesting a mediating role of OXGR1^AG in AKG-induced E release.

To further determine the role of OXGR1 in AKG's metabolic effects *in vivo*, we generated an OXGR1 global knockout mouse model (OXGR1KO) by using the clustered regularly interspaced short palindromic repeats (CRISPR) method (Appendix Fig S1A and B). We found OXGR1KO mice showed completely abolished OXGR1 mRNA expression compared to WT control mice (Appendix Fig S1C), which validates our knockout model. Surprisingly, we failed to found any metabolic phenotypes in OXGR1KO mice when fed on chow (Appendix Fig S1D–O). However, consistent with *in vitro* chromaffin cell model, when HFD-fed mice were supplemented with AKG, OXGR1KO abolished AKG-induced release of serum E (Appendix Fig S2A) as well as the inhibitory effects of AKG on HFD-induced increases in body weight (Fig 5H) and fat mass (Fig 5J and K), specifically iWAT (Fig 5L) and gWAT (Fig 5M). Similarly, OXGR1KO also diminished AKG-induced inhibition on adipocyte size in both gWAT and iWAT (Appendix Fig S2C and D) and stimulation on lean mass (Fig 5J and K). These results suggest an essential role

of OXGR1 in the inhibitory effects of AKG on HFD-induced obesity.

The key mediating role of OXGR1 is further supported by our results from the indirect calorimetry system. We found that while AKG did not change food intake in both WT and OXGR1KO mice (Fig 5I), AKG increased oxygen consumption and decreased RER in WT but not OXGR1KO mice (Fig 5P–S). These results suggest that OXGR1 mediates the stimulatory effects of AKG on energy expenditure and fat burning. Consistently, in WT but not OXGR1KO mice, AKG increased ATGL protein in the gWAT (Fig 5N and O), the phosphorylation of HSL in both iWAT and gWAT (Fig 5N and O and Appendix Fig S2E and F), and UCP1 protein in the BAT (Appendix Fig S2B). Thus, these data further provide *in vivo* evidence to support that OXGR1 is a key mediator for the AKG supplementation-induced lipolysis and thermogenesis.

To assess if OXGR1^AG is sufficient to mediate the anti-obesity effects of AKG, we generated an OXGR1 adrenal-selective reexpression (OXGR1RE^AG) mouse model by delivering HBAAV2/9-OXGR1 virus into the adrenal gland of OXGR1KO mice. In this model, adrenal OXGR1 expression and the stimulatory effects of AKG on serum E levels were both successfully rescued compared to control OXGR1KO mice (Appendix Fig S2G and H), suggesting an OXGR1^AG-mediated E-releasing effect of AKG. AKG showed no effects on food intake in both OXGR1KO and OXGR1RE^AG mice (Fig 6B). Similar to what we observed before, in the control OXGR1KO mice, AKG failed to induce metabolic phenotypes (Fig 6A–L). On the other hand, we showed similar anti-obesity effects of AKG in OXGR1RE^AG mice as what we observed in WT mice. These AKG-induced anti-obesity effects include decreases in body weight gain (Fig 6A), fat mass (Fig 6C and D), gWAT and iWAT weight (Fig 6E and F), adipocyte size of gWAT and iWAT (Appendix Fig S2J and K), and RER (Fig 6K and L), as well as increases in lean mass (Fig 6C and D), oxygen consumption (Fig 6I and J), ATGL protein in gWAT (Fig 6G and H), phosphorylation of HSL in both iWAT and gWAT (Fig 6G and H, Appendix Fig S2L and M), and UCP1 protein in BAT (Appendix Fig S2I). These results indicate OXGR1^AG is sufficient to mediate the anti-obesity effects of AKG.

Consistent with this point of view, we found enhanced anti-obesity effects of AKG in the OXGR1 adrenal-specific overexpression mouse model (GRP99OE^AG). In this model, the HBAAV2/9-OXGR1 virus was delivered into the adrenal gland of WT mice. The mRNA of OXGR1 was successfully overexpressed in the adrenal gland of GRP99OE^AG mice compared to WT mice injected with HBAAV2/9-GFP control virus (Appendix Fig S3A). We found that adrenal overexpression of OXGR1 enhanced the anti-obesity effects of AKG. For example, GRP99OE^AG enhanced AKG's inhibitory effects on body weight gain (Appendix Fig S3B), fat mass (Appendix Fig S3D and E), gWAT and iWAT weight (Appendix Fig S3F and G), adipocyte size of gWAT and iWAT (Appendix Fig S3P and Q), and RER (Appendix Fig S3N and O), as well as stimulatory effects on serum E levels (Appendix Fig S3K), lean mass (Appendix Fig S3D and E), oxygen consumption (Appendix Fig S3L and M), ATGL protein in gWAT (Appendix Fig S3H and I), phosphorylation of HSL in both iWAT and gWAT (Appendix Fig S3H and I, Appendix Fig S3R and S), and UCP1 protein in BAT (Appendix Fig S3J). Additionally, AKG showed no

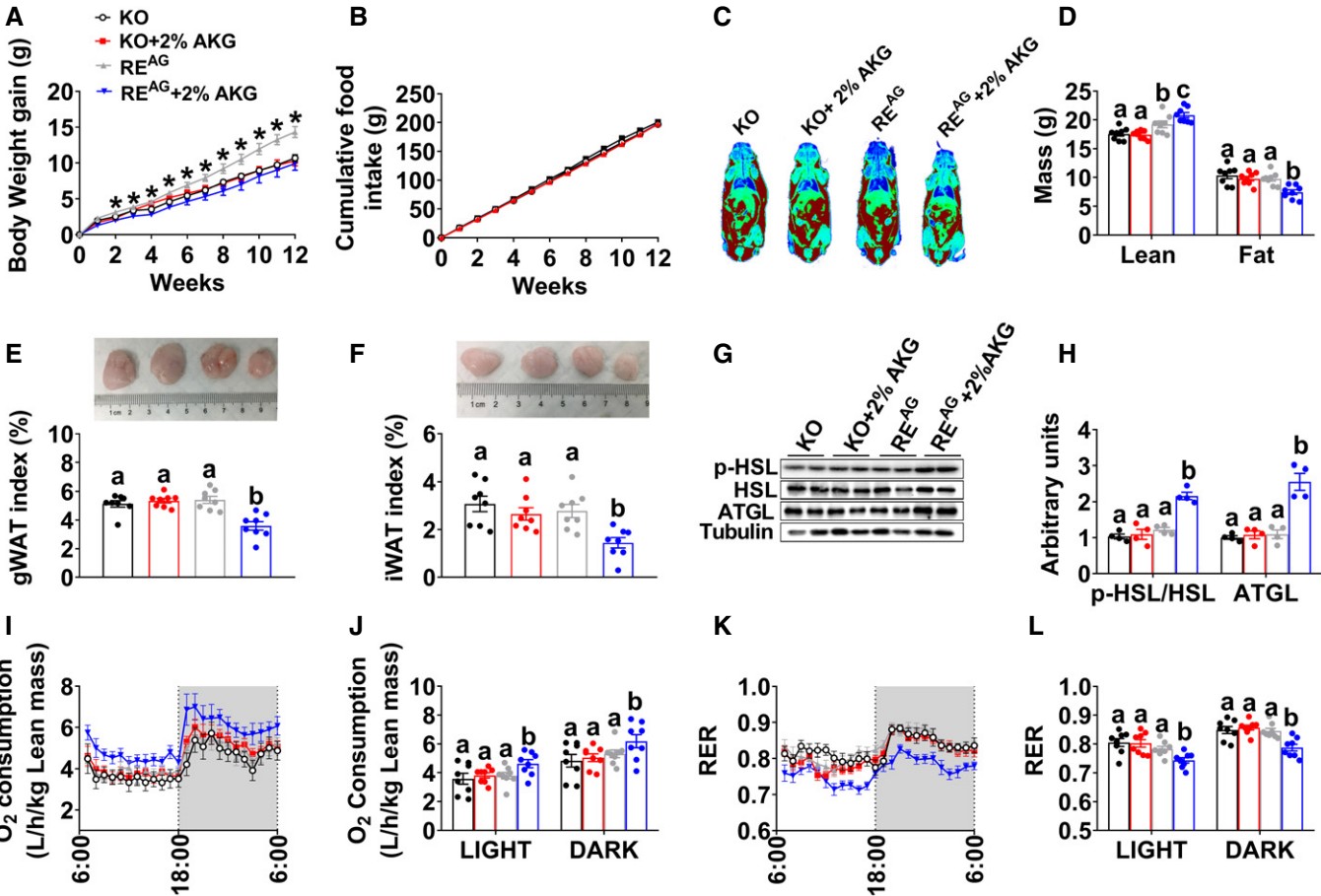

**Figure 6. OXGR1 expressed by adrenal gland mediates the stimulatory effects of AKG on thermogenesis and lipolysis.**

A, B   Body weight gain (A) and cumulative food intake (B) of male OXGR1 adrenal-specific reexpression mice (OXGR1RE^AG). Male OXGR1KO mice (8 weeks old) were adrenal-specifically injected with control HBAAV2/9-GFP (OXGR1KO control) or HBAAV2/9-OXGR1 (OXGR1RE^AG). Two weeks after injections, mice were switched to HFD and further divided into two groups receiving tap water or water supplemented with 2% AKG for 12 weeks (*n* = 8 per group).

C, D   Representative image of body composition (C) and fat and lean mass index (D) of male OXGR1RE^AG mice treated with AKG for 12 weeks (*n* = 8 per group).

E, F   Weight index of gWAT (E) and iWAT (F) in male OXGR1RE^AG mice treated with AKG for 12 weeks (*n* = 8 per group).

G, H   Immunoblots (G) and quantification (H) of p-HSL and ATGL protein in gWAT of male OXGR1RE^AG mice treated with AKG for 12 weeks (*n* = 8 per group).

I–L   Oxygen consumption (I–J) and RER (K–L) of male OXGR1RE^AG mice treated with AKG for 12 weeks (*n* = 8 per group).

Data information: Results are presented as mean ± SEM. In (A, B), *$P \leq 0.05$ by two-way ANOVA followed by post hoc Bonferroni tests. In (D–F, H, J and L), different letters between bars indicate $P \leq 0.05$ by one-way ANOVA followed by post hoc Tukey's tests.

effect on food intake in both WT control and GRP99OE^AG mice (Appendix Fig S3C). Thus, the results from both loss-of-function and gain-of-function models demonstrate that OXGR1^AG mediates the AKG supplementation-induced adipose tissue thermogenesis and lipolysis, in turn preventing DIO.

**OXGR1 is required for beneficial metabolic effects of exercise**

Our data suggest that exercise increases AKG and OXGR1 mediates anti-obesity effects of AKG supplementation. Based on these observations, we tested whether OXGR1 is required for exercise-induced beneficial metabolic effects by comparing the salutary effects of 2-week resistance exercise in OXGR1KO and WT mice. We found that resistance exercise significantly increased serum AKG level in WT mice, but this stimulatory effect was attenuated in OXGR1KO mice (Fig 7H). This attenuation may be resulted

from the reduction of total muscle mass induced by OXGR1KO when mice were fed on HFD (Figs 5K and 7G). We found exercise did not change food intake in both WT and OXGR1KO mice (Fig 7D), consistent with the previous observations in the adult and aged male mice following the same ladder-climbing resistance exercise (Kim *et al*, 2016). Exercise effectively decreased body weight gain in WT control mice, while deletion of OXGR1 attenuated the body weight decrease induced by resistance exercise (Fig 7A and B). Although resistance exercise significantly decreased fat mass (Fig 7G), specifically gWAT and iWAT weight (Fig 7E and F), in both OXGR1KO and WT control mice, these resistance exercise-induced decreases were higher in WT than OXGR1KO mice (Fig 7C and E–G). Similar attenuations were found in resistance exercise-induced increases in lean mass (Fig 7G), serum E levels (Fig 7I), UCP1 mRNA expression (Fig 7J) and protein expression (Fig 7K) in BAT, ATGL and HSL mRNA

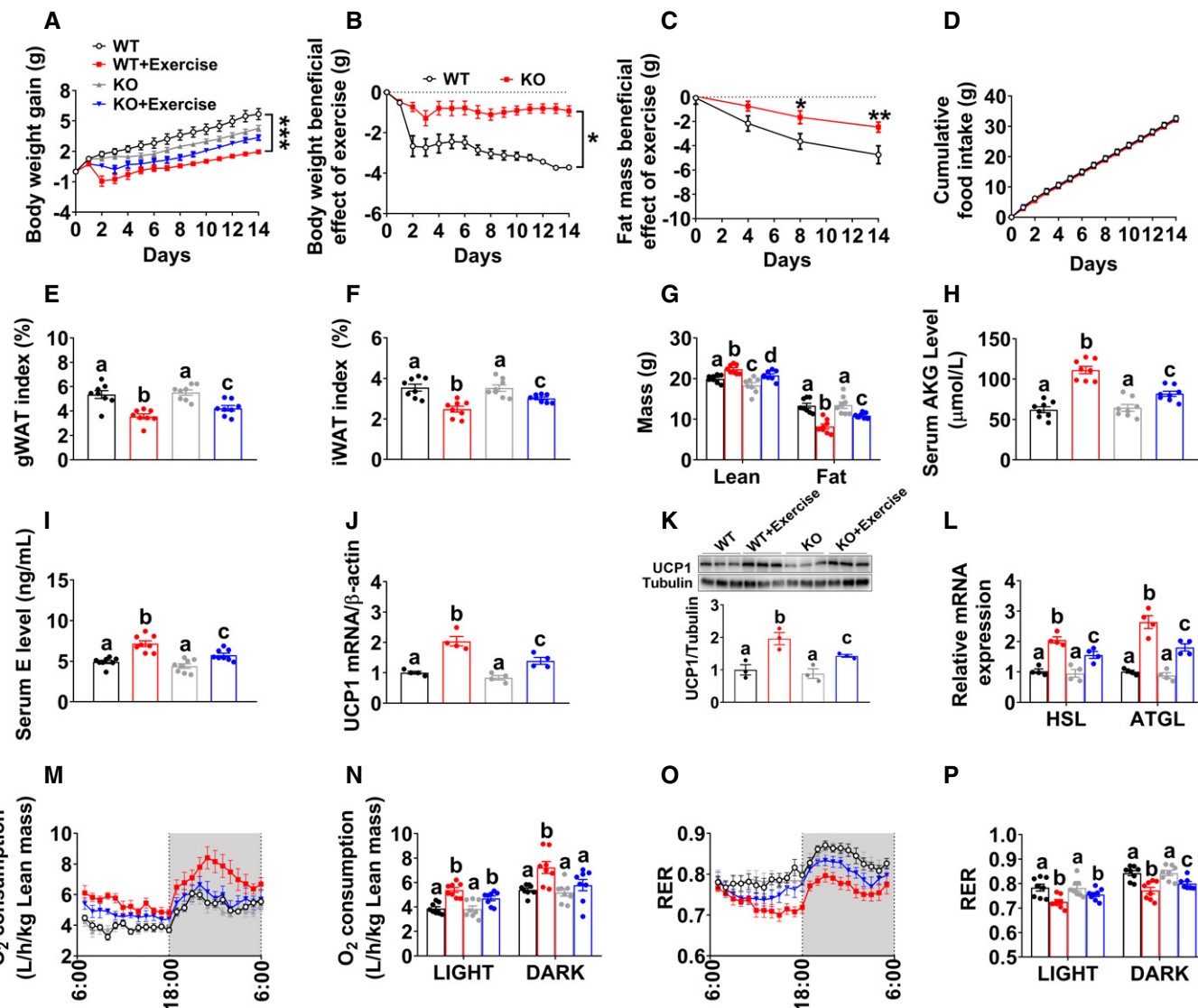

**Figure 7. OXGR1 is required for metabolic beneficial effects of resistance exercise.**

A  Body weight gain in male WT littermates and OXGR1KO mice. At 8 weeks of age, male C57BL/6 WT control or OXGR1KO mice were switched to HFD. After 12 weeks of HFD feeding, mice were further divided into two groups receiving non-exercise or resistance exercise for 14 days (*n* = 8 per group).

B  Exercise-induced body weight loss in male WT littermates and OXGR1KO mice. Body weights from exercise mice were subtracted by the average body weight of the non-exercise control group for each genotype (*n* = 8 per group).

C  Exercise-induced fat mass loss in male WT littermates and OXGR1KO mice. Fat mass from exercise mice was subtracted by the average fat mass of the non-exercise control group for each genotype (*n* = 8 per group).

D  Cumulative food intake of male WT littermates and OXGR1KO mice after 14-day resistance exercise (*n* = 8 per group).

E, F  Weight index of gWAT (E) and iWAT (F) of male OXGR1KO mice after 14-day resistance exercise (*n* = 8 per group).

G  Body composition of male OXGR1KO mice after 14-day resistance exercise (*n* = 8 per group).

H  Serum AKG levels of male WT and OXGR1KO mice after resistance exercise. Male WT and OXGR1KO mice (10 weeks old) fed with normal chow were receiving resistance exercise for 40 min (*n* = 8 per group). The serum AKG levels were tested before and immediately after exercise.

I  Serum E level in male OXGR1KO mice after 14-day resistance exercise (*n* = 8 per group).

J–L  The mRNA expression (J) and protein expression of UCP1 (K) in the BAT or the mRNA expression of HSL and ATGL (L) in the gWAT of male OXGR1KO mice after 14-day resistance exercise (*n* = 4 per group).

M–P  Oxygen consumption (M, N) and RER (O, P) in male OXGR1KO mice after 14-day resistance exercise (*n* = 8 per group).

Data information: Results are presented as mean ± SEM. In (A–D) *$P \le 0.05$ and **$P \le 0.01$ by two-way ANOVA followed by post hoc Bonferroni tests. In (E–L, N and P), different letters between bars indicate $P \le 0.05$ by one-way ANOVA followed by post hoc Tukey's tests.

expression in gWAT (Fig 7L), and oxygen consumption (Fig 7M and N), as well as decreases in RER (Fig 7O and P). These data collectively support a model in which resistance exercise increases AKG secretion from muscle tissues to bind with adrenal OXGR1 and, by doing so, increases adipose tissue lipolysis and thermogenesis and prevents DIO.

**The p65/NF-κB inflammatory pathway is required for the stimulatory effects of AKG on E release from chromaffin cells *in vitro***

To explore the intracellular mechanism of AKG-induced E release, we investigated the transcriptomic alteration induced by AKG treatment in *in vitro* adrenal chromaffin cells by RNA sequencing. Ingenuity pathway analysis (IPA) was used for functional annotation of the genes differentially expressed between the control and AKG treatment groups. Unexpectedly, we found that AKG activated the inflammatory responses, especially the cytokine interleukin (IL) pathways (Fig 8A). It is well known that the expression of inflammatory genes and pro-inflammatory cytokines is mainly regulated by nuclear factor kappa B (NF-κB) family of transcription factors (Karin *et al*, 2004; Hu *et al*, 2005). Here, we showed that AKG effectively increased the phosphorylation of IκB kinase (IKK, an upstream activator for NF-κB) and nuclear factor of kappa light polypeptide gene enhancer in B-cells inhibitor, alpha (IκBα, an inhibitor of NF-κB; Fig 8B and C), suggesting an AKG-induced NF-κB activation. We speculate that AKG activates IKK to phosphorylate the inhibitory IκBα protein, which leads to the dissociation of IκBα from NF-κB and subsequent nuclear shuttling and activation of NF-κB. In support of this view, we found AKG increased the expression of p65, a subunit of NF-κB transcription complex, in the nucleus protein extraction, while it decreased p65 in the cytoplasm (Fig 8B and C). The same nucleus NF-κB (p65) translocation was also observed by IF staining (Fig 8D). These results indicated that AKG activates NF-κB inflammatory pathway in *in vitro* adrenal chromaffin cells.

To directly test if OXGR1 is required for the stimulatory effects of AKG on the NF-κB pathway, an OXGR1-knockdown chromaffin cell model was generated by using siOXGR1. We found OXGR1 knockdown significantly decreased protein expression of OXGR1 (Fig 8B and C), which validated our loss-of-function model. By using this model, we showed that the knockdown of OXGR1 abolished the stimulatory effects of AKG on OXGR1 protein expression and NF-κB signal transduction cascade (Fig 8B–D), suggesting a mediating role of OXGR1 in AKG-induced NF-κB signaling activation. It has been previously shown that circulating cytokines affect chromaffin cell secretory function through NF-κB activation (Ait-Ali *et al*, 2008; Douglas *et al*, 2010; Bunn *et al*, 2012). NF-κB signaling activation may play a role in AKG-induced E secretion. Consistent with this speculation, we found that while AKG treatment increased the release of E and activated NF-κB signaling, both stimulatory effects were abolished by co-treatment with IKK16, an IKK inhibitor (Fig 8E–G). These results suggest a mediating role of NF-κB activation in the stimulatory effects of AKG on E release *in vitro*.

## Discussion

The major finding of our study is that exercise-responsive myometabolite, AKG, prevents HFD-induced body weight gain and adiposity in both male and female mice. Systemic characterization revealed normal food intake, but robust increases in energy expenditure, BAT thermogenesis, and WAT lipolysis induced by AKG. We further provided both *in vitro* and *in vivo* evidence supporting that the anti-obesity effect of AKG is mediated by adrenergic stimulation of adipose tissue thermogenesis and lipolysis. By using both loss-of-

function and gain-of-function mouse models, we showed that AKG receptor OXGR1, expressed by adrenal glands, is essential for the stimulatory effects of AKG on thermogenesis and lipolysis. Importantly, we demonstrated that OXGR1 is required for exercise-induced weight loss and fat reduction. We also provided *in vitro* evidence supporting that the adrenal activation of p65/NF-κB inflammatory pathway is required for the stimulatory effects of AKG on E release. These findings implicate myometabolite AKG in the physiological mechanism underlying exercise-induced weight loss and demonstrate that AKG acts as a previously unappreciated systemic adrenergic signal and exerts profound effects on whole-body metabolism.

The comprehensive serum metabolite signatures induced by acute resistance exercise include decreased amino acids and increased fatty acids. These findings are consistent with previous observations that the oxidation and catabolism of amino acids, especially branched-chain amino acids (BCAA), are promoted by exercise (Qun *et al*, 2014), while the mobilization of free fatty acids from depots and efflux of plasma free fatty acids are increased by exercise (Friedberg *et al*, 1963; Shimomura *et al*, 2004). Importantly, the metabolites of valine, i.e., alpha-ketoisovaleric acid (α-keval) and 2-hydroxy-3-methylbutyric acid (2H3MA), as well as a metabolite of alanine, pyruvic acid (Pyr), were also increased after acute exercise. All these three metabolites can be converted into acetyl-CoA or succinyl-CoA, which is the main input or important intermediate for TCA oxidation (Li *et al*, 2017). Consistently, AKG, another essential intermediate in the TCA cycle, was found to be upregulated by exercise.

We successfully identified several rapid response metabolites (pyruvate, lactate, malate, succinate, AKG, xanthine, and hypoxanthine) induced by resistance exercise. In line with other literature (Yde *et al*, 2013; Berton *et al*, 2017), we found resistance exercise induced a rapid accumulation of pyruvate and lactate, reflecting anaerobic metabolism and muscle damage (Gorostiaga *et al*, 2014). The increase in pyruvate is due to the limited ability of mitochondria to oxidase pyruvate during anaerobic exercise. To fulfill the high energy demand required by resistance exercise, pyruvate is converted to lactate in muscle and then transported through the bloodstream to the liver, where lactate can be converted into glucose by gluconeogenesis. The increased levels of pyruvate and lactate validate our resistance exercise model.

Interestingly, opposite to well-established accumulation signatures of malate, succinate, hypoxanthine, and xanthine following endurance exercise (Lewis *et al*, 2010; Aguer *et al*, 2017), our metabolomics analyses found these metabolites decreased following acute resistance exercise. The same trend was consistently demonstrated by LC–MS/MS analyses comparing the metabolic response following endurance and resistance exercise, suggesting an exercise type-specific metabolic response. Notably, succinate was found to be increased shortly after resistance exercise (bilateral leg extension exercises) in humans (Berton *et al*, 2017). This discrepancy may be attributed to the different forms of resistance exercise (i.e., bilateral leg extension versus ladder-climbing), the time points taken into consideration in the studies (i.e., 5 min after versus immediately after), and the research subjects (i.e., moderately trained humans versus untrained mice).

Consistent with a previous report (Leibowitz *et al*, 2012), we found AKG significantly increased in the blood following resistance

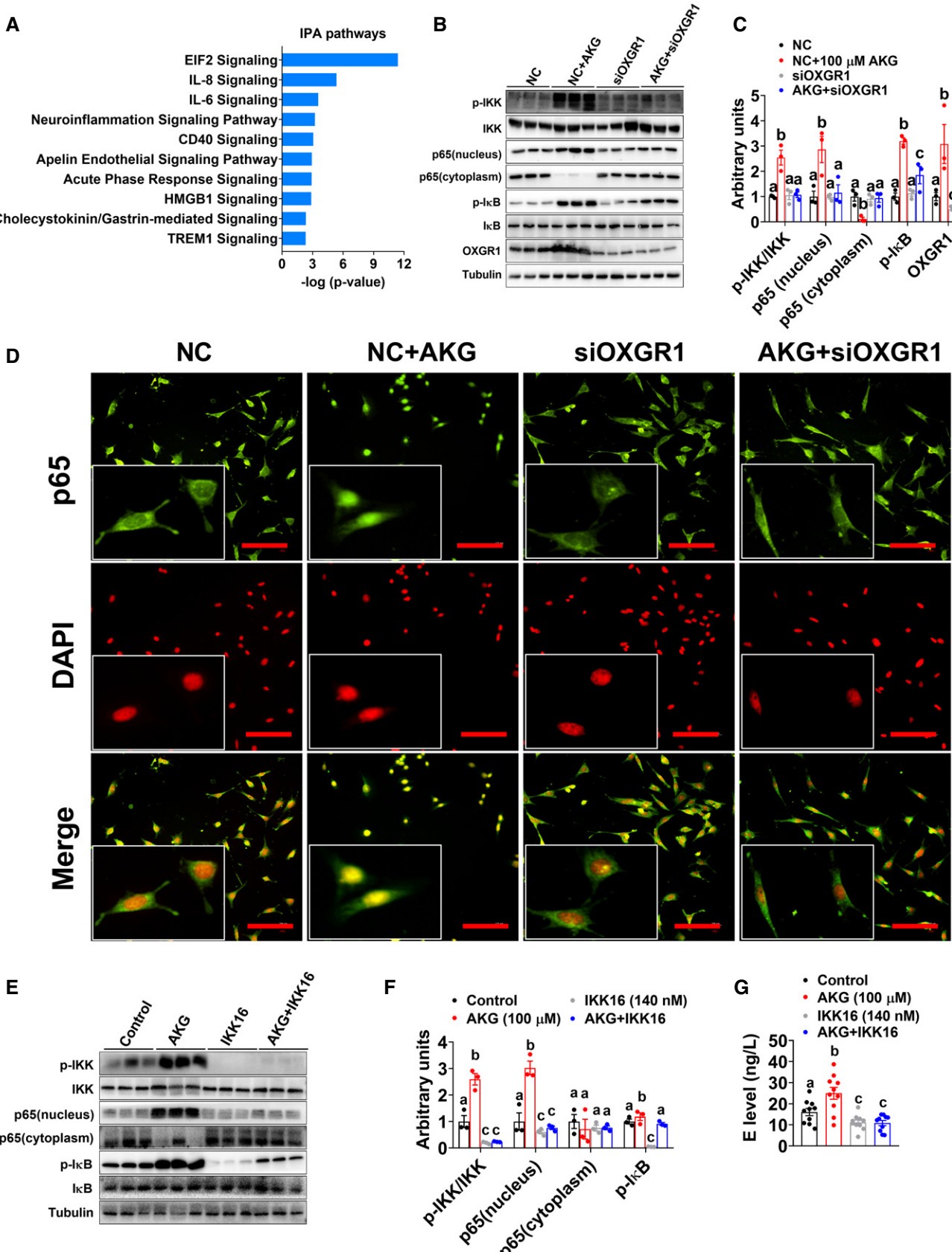

Figure 8.

◄ **Figure 8.  p65/NF-κB inflammatory pathway is required for the stimulatory effects of AKG on E release of adrenal chromaffin cells *in vitro*.**

A    Ingenuity pathway analysis (IPA) of AKG-induced transcriptome signature in adrenal chromaffin cells treated with AKG. The mRNA was extracted from adrenal chromaffin cells after 3-h incubation of vehicle or AKG (100 μM) (*n* = 3 per group).

B, C    Immunoblots (B) and quantification (C) of p-IKK/IKK, p65, p-IκB/IκB, and OXGR1 protein in adrenal chromaffin cells cultured with vehicle + NC, vehicle + siOXGR1, AKG (100 μM) + NC, or AKG + siOXGR1 for 3 h (*n* = 3 per group).

D    p65 translocation in adrenal chromaffin cells cultured with vehicle + NC, vehicle + siOXGR1, AKG (100 μM) + NC, or AKG + siOXGR1 for 3 h (*n* = 3 per group). Scale bars, 100 μm.

E, F    Immunoblots (E) and quantification (F) of p-IKK/IKK, p65, and p-IκB/IκB protein in adrenal chromaffin cells cultured with vehicle, AKG (100 μM), IKK inhibitor IKK16, or AKG + IKK16 for 3 h.

G    E level in the medium from adrenal chromaffin cells cultured with vehicle, AKG (100 μM), IKK16, or AKG + IKK16 for 3 h (*n* = 10 per group).

Data information: Results are presented as mean ± SEM. In (C, F and G), different letters between bars indicate $P \leq 0.05$ by one-way ANOVA followed by post hoc Tukey's tests.

exercise. Interestingly, we also found resistance exercise decreased circulating glutamate and leucine, both of which can be metabolized into AKG. The observed elevation in AKG levels may be attributed to glutamate and leucine degradation. Consistent with this point of view, both leucine degradation (Pechlivanis *et al*, 2010) and glutamate breakdown (Leibowitz *et al*, 2012) were found to increase the circulating AKG. Considering the critical role of AKG in the TCA cycle, it is likely that amino acid metabolism (leucine degradation) and muscle glutamate content depletion contribute to the energy supply during resistance exercise.

AKG is an important biological molecule with pleiotropic activity and has been shown to have broad therapeutic potentials, such as decreasing risk of cancer (Mullen *et al*, 2014), maintaining intestinal health (Hou *et al*, 2011), promoting muscle growth (Cai *et al*, 2016), and orchestrating macrophage activation through epigenetic alteration (Liu *et al*, 2017; Zdzisinska *et al*, 2017). Notably, a potential role of AKG in energy homeostasis has also been suggested by our observations. When fed on chow, AKG-treated mice showed upregulation of lean muscle mass and body weight gain. These results are consistent with our previous observations that AKG increases muscle protein synthesis (Cai *et al*, 2016) while decreasing muscle protein degradation (Cai *et al*, 2018). On the other hand, we found AKG treatment increases BAT thermogenesis and decreases fat mass, which is consistent with the previous report that AKG increases BAT adipogenesis and thermogenesis via an epigenetic way (Yang *et al*, 2016). These metabolic changes induced by AKG resemble several key metabolic responses induced by resistance exercise, i.e., enhanced thermogenesis, increased muscle mass, and decreased fat pads (Allen *et al*, 2001; Stanford & Goodyear, 2016). Additionally, oral administration of AKG has been previously shown to decrease adiposity in a DIO rat model (Tekwe *et al*, 2012). Based on these observations, we postulated that AKG might have similar beneficial metabolic effects on DIO as resistance exercise.

Consistent with this, we found AKG prevented body weight gain induced by HFD, which is associated with increased energy expenditure but not food intake. Notably, decreased body weight gain is due to a superior portion of fat mass loss compared to lean mass gain. As we observed in chow-fed mice, AKG increased BAT expression of thermogenic genes including UCP1, Dio2, and Cidea, suggesting upregulation of BAT thermogenesis. Similar AKG-induced WAT lipolysis was also indicated by upregulated RER and increased expression of ATGL and p-HSL, the main enzymes catalyzing lipolysis in WAT (Bolsoni-Lopes & Alonso-Vale, 2015). These results suggest that the water supplement of AKG prevents DIO by increasing BAT thermogenesis and WAT lipolysis.

Exercise-induced myometabolites cause physiological changes in target tissues either directly or indirectly by affecting the secretion of endogenous hormones. As an exercise-induced metabolite (Leibowitz *et al*, 2012), AKG has been shown to exert direct regulatory effects on the muscle development (Cai *et al*, 2016), liver injury and repair (Wang *et al*, 2015), and intestinal immune protection (Hou *et al*, 2011). It is unknown whether AKG directly acts on the BAT or indirectly acts through other tissues to increase thermogenesis. To test the direct effects of AKG on BAT, we used both *ex vivo* BAT and *in vitro* primary brown adipocyte culture models. We found that direct treatment with AKG failed to affect dissolved oxygen, NEFA levels, or calcium signaling in both models, suggesting an alternative indirect effect of AKG. Importantly, AKG supplementation has been shown to mediate the stimulatory effects of dietary restriction on lifespan by inhibiting ATP synthase and rapamycin (TOR) signaling (Chin *et al*, 2014). It is possible that AKG acts on ATP synthase and TOR signaling to regulate BAT thermogenesis. However, AKG treatment also failed to regulate ATP production or protein expression of p-AMPKα and p-FoxO1. These results indicate an indirect regulatory role of AKG in BAT metabolism.

Both BAT thermogenesis and WAT lipolysis are under coordinated control by metabolic hormones. For example, thyroid hormones T4 and its active form, T3, as well as type 2 deiodinase (D2), an essential enzyme activating T4 to T3, are required for adaptive thermogenesis in BAT (de Jesus *et al*, 2001; Mullur *et al*, 2014). E and NE, the catecholamines secreted from the adrenal glands, have been shown to activate triglyceride lipase and induce lipolysis in WAT (Jocken & Blaak, 2008; Bartness *et al*, 2014), and also upregulate UCP1 and stimulate BAT thermogenesis (Collins *et al*, 2010; Sharara-Chami *et al*, 2010). Here, we found that AKG increased the mRNA expression of Dio2, the gene coding D2 protein, in the BAT, suggesting an increase in conversion from T4 to T3. However, we failed to observe any changes in serum T3 and T4 levels.

Interestingly, we found AKG significantly stimulated the release of E, the main hormone secreted by the adrenal medulla. Considering the stimulatory effects of E on both BAT thermogenesis and WAT lipolysis, it is likely that AKG promoted BAT thermogenesis and WAT lipolysis through stimulating the release of E from the adrenal medulla. In supporting this view, we found that the acute stimulatory effects of AKG on energy expenditure and fat burning (indicated by oxygen consumption and RER) were abolished by the systemic blockage of ADRB3. More importantly, the adrenalectomized male mice showed no metabolic responses to water supplementation of AKG. These findings indicate an indirect stimulation of

AKG on adipose tissue thermogenesis and lipolysis through adrenal released E.

While activation of the BAT thermogenesis has clinically significant effects, BAT is found in negligible volumes in adult humans (Yoneshiro *et al*, 2013). Indeed, cold exposure does not induce measurable metabolic response of small BAT depots in humans (Chondronikola *et al*, 2014). Instead, data from rodents and humans (Wu *et al*, 2012; Betz & Enerback, 2015) suggest BAT-like white adipose tissue (beige or brite) as an intriguing and potentially more physiologically significant target for human metabolic syndrome. Indeed, human pre-adipocytes can be differentiated into brite adipocytes, which resemble classical brown adipocytes and respond to cold adaptation or other stimuli (Giralt & Villarroya, 2013; Rosenwald & Wolfrum, 2014; Betz & Enerback, 2015). Interestingly, increased circulating E levels have been shown to enhance britening in the subcutaneous white adipose tissue in both mice and humans (Cannon & Nedergaard, 2004; Kajimura *et al*, 2015; Sidossis *et al*, 2015). Consistently, we found that water supplementation of AKG significantly increased brite adipocytes that expressed UCP1 and other brite and thermogenic genes in the iWAT. These data suggest that britening, in rodents, may have therapeutic relevance, which can be enhanced by AKG supplementation.

One important issue to be considered is the direct effects of E on the behavior of the mice. Consistent with the well-established functions of circulating E (Tank & Lee Wong, 2015), we found that acute AKG treatment increased both heart rate and blood pressure but not physical activity. These cardiac changes are consistent with the increased demand for higher blood glucose and free fatty acids during exercise, suggesting a possible role of AKG in exercise. The release of E from the adrenal gland is tightly controlled by SNS (Grassi & Ram, 2016), and AKG may increase the adrenal release of E by increasing SNS input to the adrenal gland. However, we found AKG treatment increases PLCβ protein, and releases of calcium ion and E from *ex vivo* adrenal gland and *in vitro* adrenal medullary chromaffin cells. These findings suggest that AKG may directly act on the adrenal gland to increase instantaneous intracellular calcium and, by doing so, promote the release of E.

To investigate how AKG interacts with the adrenal medulla to increase the release of E, we examined adrenal expression levels of different AKG sensors, which have been previously shown to interact with AKG to exert physiological functions. These include the classical sensors of AKG, i.e., JMJDs, TETs, PHDs, and GPRs. Among these different AKG sensors, OXGR1, a verified AKG receptor, was found to have the highest expression in the adrenal gland. OXGR1 is an orphan G protein-coupled receptor first discovered in 2002, and later was identified as the receptor for AKG and renamed as 2-oxoglutarate receptor 1 (Wittenberger *et al*, 2002; He *et al*, 2004). It has been shown that the half-maximal effective concentration (EC50) for OXGR1 to AKG is ~70 µM (He *et al*, 2004), which is equal to the concentration of circulating AKG at rest condition (69.8 ± 2.8 µM; Fig 1H), suggesting a baseline activation of OXGR1. Notably, both wheel-running and ladder-climbing significantly increase circulating AKG (91.3 ± 3.7 µM versus 112.2 ± 3.2 µM; Fig 1I), indicating an enhanced OXGR1 activation induced by exercise. Previous studies have indicated that OXGR1 plays an important role in mucin regulation in otitis (Kerschner *et al*, 2013) and cardiac hypertrophy (Omede *et al*, 2016). However, there is no known role of OXGR1's effects on fat thermogenesis and

lipolysis. We found that while OXGR1 is widely expressed in many tissues, adrenal glands, testes, and brain have the highest expression of GRP99, which is consistent with a previous report (Diehl *et al*, 2016). Importantly, OXGR1 is highly expressed inside the adrenal gland medulla but not adrenal cortex, the main region that releases E. These results suggest a possible mediating role of OXGR1 in the stimulatory effects of AKG on E release.

To directly test if OXGR1 mediates the anti-obesity effects of AKG, we used CRISPR gene-editing technology to generate a global OXGR1KO mouse line. The single-guide RNA (sgRNA) was designed to target the exon 4 of OXGR1 locus, and OXGR1 expression is effectively disrupted in the OXGR1KO model. We found AKG showed no effects on serum E levels in OXGR1KO mice. Interestingly, AKG also failed to promote lipolysis or prevent DIO in these OXGR1KO mice, suggesting a key role of OXGR1 in mediating the metabolic effects of AKG. These attenuations are not a result of CRISPR-mediated off-site mutagenesis, as virus-mediated selective reexpression of OXGR1 in the adrenal gland of OXGR1KO mice rescued AKG's effects on body weight, energy expenditure, fat thermogenesis, and lipolysis. Consistently, selective overexpression of OXGR1 in the adrenal gland of WT mice enhanced these effects of AKG. These findings demonstrate that OXGR1 expressed in the adrenal gland has a major role in the anti-obesity effects of AKG. However, we cannot exclude the potential roles of OXGR1 expressed in other tissues. There is a possibility that AKG acts through the central nervous system (CNS) to increase sympathetic input to fat tissue and, by doing so, to prevent DIO. Consistent with this view is that OXGR1 has high mRNA expression in the hypothalamus, which has been reported to regulate fat thermogenesis and lipolysis through SNS (Contreras *et al*, 2017). Interestingly, in our OXGR1RE[AG] model, adrenal reexpression of OXGR1 cannot fully rescue the anti-obesity effects of AKG, suggesting that other OXGR1 pathways may be involved. Our laboratory is currently exploring if the metabolic effects of AKG are partially mediated by central OXGR1.

Although the beneficial metabolic effects of AKG supplementation were blocked in HFD-fed OXGR1KO mice, we failed to observe any metabolic phenotypes of OXGR1KO mice when fed on chow diet. These results suggest that OXGR1 is not required for energy homeostasis regulation and body weight control on chow when circulating AKG is at a baseline level (no AKG supplementation or exercise). It appears that OXGR1 only exerts beneficial effects on metabolic health and particularly muscle hypertrophy and fat loss when circulating AKG is high. Interestingly, our data indicate that resistance exercise effectively increases serum AKG level in both chow- and HFD-fed mice. These raise the possibility that AKG/OXGR1 signaling mediates the salutary effects of anaerobic exercise. Consistent with this view, we found exercise-induced metabolic beneficial effects, including body weight loss, lipolysis, and fat mass reduction, were largely attenuated in OXGR1KO mice. Importantly, in OXGR1KO mice, resistance exercise effectively increased serum AKG up to the level we observed after AKG water supplementation, while baseline AKG level is still normal, suggesting a normal AKG response induced by resistance exercise. Taking all these together, our data support a key role of AKG/OXGR1 signaling in beneficial metabolic effects induced by resistance exercise.

Another interesting phenotype we observed in OXGR1KO mice is muscle hypotrophy and body weight loss when fed on HFD, suggesting a potential protective effect of OXGR1 on HFD-induced

muscle loss. Notably, we showed that HFD significantly decreased baseline circulating AKG level in mice and serum AKG level is negatively corrected with BMI in humans. So OXGR1 protects HFD-induced muscle loss when circulating AKG is low, which is independent of resistance exercise. The protective effect of OXGR1 is possibly mediated by directly increasing muscle protein synthesis (Cai *et al*, 2016) and decreasing muscle protein degradation (Cai *et al*, 2018). Of course, it is also possible that other indirect pathways are involved. DIO has been shown to lead to skeletal muscle atrophy, associated with upregulation of muscle-specific ubiquitin ligases, oxidative stress, myonuclear apoptosis, and autophagy (Abrigo *et al*, 2016). So OXGR1-mediated protection is also possible due to the compensatory response to the other physiological changes induced by OXGR1KO, e.g., cardiac hypertrophy (Omede *et al*, 2016) or impaired acid–base homeostasis (Tokonami *et al*, 2013).

As the first step to explore the intracellular mechanism of AKG-induced E release, we performed transcriptome analysis on adrenal chromaffin cells before or after AKG treatment. Interestingly, inflammatory signaling pathways were identified as the transcriptome signature induced by AKG treatment. We further provided *in vitro* evidence to support a model in which AKG acts through OXGR1 to induce NF-κB signal transduction cascade, and by doing so, it stimulates E release from chromaffin cells. Consistent with this model, pro-inflammatory cytokines tumor necrosis factor-α, and interleukin-1β/α have been shown to modulate catecholamine secretion and long-term gene regulation in chromaffin cells of adrenal medulla (Rosmaninho-Salgado *et al*, 2009; Tamura *et al*, 2014; Jenkins *et al*, 2016). The activation of a cytokine-specific intracellular signaling pathway and following E2 release in adrenal medulla may be one part of a systemic stress response induced by exercise. Chromaffin cells may act as an integrative center responding to both immune and endocrine stimuli induced by exercise. Our laboratory is currently validating this model and exploring if p65/NF-κB inflammatory pathway is required for the stimulatory effects of AKG on E release of adrenal chromaffin cells *in vivo*.

In conclusion, we found that exercise-induced metabolic benefits are mediated through a systemic increase in the TCA cycle intermediate AKG. AKG exerts acute and chronic control over adipose tissue thermogenesis and lipolysis by stimulating the release of E through OXGR1, an AKG sensor expressed by the adrenal gland. Thus, our findings identify AKG as an exercise-responsive myometabolite with essential salutary metabolic effects, acting as a previously unappreciated systemic pathway for activation of adipocyte thermogenesis and lipolysis.

# Materials and Methods

### Animals

Mice were housed in a temperature-/humidity-controlled environment (23°C ± 3°C/70% ± 10%) on a 12-h light/12-h dark cycle (6 am and 6 pm). Unless otherwise stated, the mice were maintained *ad libitum* on standard mouse chow (Protein 18.0%, Fat 4.5%, and Carbohydrate 58%; Guangdong Medical Science Experiment Center, Guangzhou, Guangdong, China) and water. All groups within one experiment contain individual mouse with the same strain and sex, showing similar body weight and age. All used mice aged between

10 and 20 weeks at the time when they were sacrificed. C57BL/6 mice were purchased from the Animal Experiment Center of Guangdong Province (Guangzhou, Guangdong, China). C57BL/6 mice were used for acute or long-term experiments to study AKG's metabolic effects. The OXGR1KO mice (Shanghai Research Center for Model Organisms, Shanghai, China) were generated and maintained on a C57BL/6 background. They were used to study the metabolic effects of long-term AKG supplementation. Care of all animals and procedures in South China Agricultural University conformed to "The Instructive Notions with Respect to Caring for Laboratory Animals" issued by the Ministry of Science and Technology of the People's Republic of China and were approved by the Animal Subjects Committee of South China Agricultural University.

### Adrenal chromaffin cell culture and primary tissue culture of adult adrenal gland and BAT

Mouse adrenal chromaffin cell line (cbr1301321; BIOSH Biotechnology Company, Shanghai, China) and adrenal gland or BAT obtained from 10- to 12-week-old C57BL/6 mice were cultured in high-glucose DMEM (11965175; Thermo Fisher Scientific, Carlsbad, CA, USA) at 37°C in a humidified atmosphere that contained 5% $CO_2$. The DMEM was supplemented with 10% fetal bovine serum (16000044; Thermo Fisher Scientific), 100,000 units/l of penicillin sodium, and 100 mg/l of streptomycin sulfate (11860038; Thermo Fisher Scientific).

### Metabolic signatures of exercise in mouse serum

Ten-week-old male C57BL/6 mice were divided into four groups: resistance exercise group, endurance exercise group, running-wheel exercise group, or a control group without exercise. For the resistance exercise group, resistance ladder-climbing exercise was performed as described previously (Kim *et al*, 2016). Ladder-climbing exercise was conducted by using a 1-m ladder with 1.5-cm grids. The ladder was set to attain an 80-degree angle with the ground. Three days of adaptation was conducted by letting mice to climb up the ladder without any resistance. The mice were positioned at the very bottom of a 1-meter ladder and motivated to climb up the ladder. When mice reached the very top of the ladder, a 2-min rest was given before the next trail of ladder-climbing. Mice first received resistance-training adaptations without a load attached to the tail for 1 week with 1 h of ladder-climbing per day. After the adaptation, resistance at 10% of body weight was given to the mice by adding weight on the tail. The loads were increased gradually as the exercise sessions proceeded. To progressively increase exercise intensity, 2 g of additional weight was applied after four successful trails. After about 40 min of resistance exercise, mice were exhausted and then anesthetized. Blood samples were collected by retro-orbital bleeding. The serum was separated and used for metabolomics analysis. Muscle samples, including tibialis anterior, forelimbs, gastrocnemius, pectineus, soleus, dorsal muscle, and pectoralis, were collected for AKG assay and AKG-related enzyme activity test. For the endurance exercise group, mice were forced running on a treadmill (47300 TREADMILL; Ugo Basile, Italy) as described previously (Lewis *et al*, 2010). Mice were first acclimated for 5 min a day for 3 days, at a low rate of 14 m/min and no incline. On the fourth day, the treadmill was set to a constant 10%

incline and started at 10 m/min. Every 2 min, the speed was then increased by 2 m/min, and the mice were forced to run to exhaustion. Exhaustion was determined by the unwillingness of mice to keep running on the treadmill, despite stimulus by a small electric shock on the stationary platform of the treadmill. Once determined to be exhausted, mice were euthanized. For the running-wheel exercise group, mice were singly housed and given free access to home cage running wheel for 1 day. For the control group, mice were singly housed and maintained on normal chow. Serum and muscle samples were collected from both the running-wheel and control groups as described in the resistance exercise group.

In another separate experiment, male C57BL/6 mice at 8 weeks of age were switched to HFD and continuously fed with HFD for 12 weeks to induce DIO. At 20 weeks of age, mice received about 40-min resistance exercise as described above. Serums were collected at 0, 1, 2, 4, and 6 h after exercise.

## Metabolomics analysis

Serum samples from mice receiving resistance exercise and control mice were used for metabolic signature analysis. The untargeted metabolomics profiling was performed on the XploreMET platform (Metabo-Profile, Shanghai, China) by Metabo-Profile Biotechnology Co., Ltd (Shanghai, China). The sample preparation procedures were according to the previously published methods with minor modifications (Qiu et al, 2009). Briefly, the serum samples were thawed on ice bath and centrifuged for 5 min at 4°C and 3,000 g (Microfuge 20R; Beckman Coulter, Inc., Indianapolis, IN, USA) to separate debris or a lipid layer. Each sample aliquot of 50 μl was mixed with 10 μl of internal standard and 175 μl of pre-chilled methanol/chloroform (v/v = 3/1). After incubation at 20°C for 20 min, the mixture was centrifuged at 14,000 g for 20 min at 4°C. The supernatant was transferred to an autosampler vial (Agilent Technologies, Foster City, CA, USA). All the samples in autosampler vials were evaporated briefly to remove chloroform using a CentriVap vacuum concentrator (Labconco, Kansas City, MO, USA) and further lyophilized with a FreeZone freeze dryer equipped with a stopping tray dryer (Labconco). The sample derivatization and injection were performed by a robotic multipurpose sample MPS2 with dual heads (Gerstel, Muehlheim, Germany). Briefly, the dried sample was derivatized with 50 μl of methoxyamine (20 mg/ml in pyridine) at 30°C for 2 h, followed by the addition of 50 μl of MSTFA (1% TMCS) containing FAMEs as retention indices. The mixture was further incubated at 37.5°C for another 1 h using the sample preparation head. In parallel, the derivatized samples were injected with the sample injection head after derivatization.

The GC-TOFMS raw data processing, peak deconvolution, compound annotation, statistical analysis, and pathway analysis were processed using XploreMET software (v3.0; Metabo-Profile, Shanghai, China) as described in a previous publication (Ni et al, 2016). Compound identification for GC–TOFMS was performed by comparing the mass fragments with JiaLib mass spectral databases. Principal component analysis (PCA) and orthogonal partial least squares discriminant analysis (OPLS-DA) were also performed with XploreMET. The Student's t-test was used for further differentiating variable selection and validation ($P < 0.05$). The Z-score indicates how many standard deviations an observation is above or below the mean of the control group (Fig 1D). The calculated fold change of

1.5 or P-value of 0.05 is chosen for statistical significance. The V-plot that integrates the fold change and P-values is used for depicting the significantly different metabolites (Fig 1E).

## Association between plasma AKG level and body mass index (BMI) in Chinese adults

This observational study was conducted in Zhujiang Hospital of South Medical University, between August 2018 and November 2018. A set of 45 Chinese volunteers aged from 24 to 75 (10 males and 35 females) were recruited from Zhujiang Hospital of South Medical University. One week before the start of the study, participants were asked to complete a self-administered form, including gender, age, and symptoms of heart disease and bone or joint problem. Individuals with previous history, signs, and symptoms, or self-declaration, of coronary heart disease, cardiovascular disease, and kidney disease were excluded from the study. At the study day, height and body weight were obtained with participants wearing light clothing without shoes. BMI was calculated using the equation BMI = $kg/m^2$, where kg is a participant's weight in kilograms and $m^2$ is the height in meter squared. Blood samples were collected from each participant and stored in EDTA tubes. Samples were subsequently centrifuged (4,000× g for 20 min at 4°C), and plasmas were stored in −80°C. The plasma AKG levels were measured by LC–MS/MS analysis (Uplc1290-6470A QQQ liquid chromatography–mass spectrometry instrument; Agilent Technologies). The study was reviewed and approved by the Human Subjects Ethics committee of Zhujiang Hospital of South Medical University, and written informed consent was obtained from each participant.

## Long-term effects of AKG on energy homeostasis

To investigate the metabolic effects of AKG when mice fed on chow, both male and female C57BL/6 mice at 12 weeks of age were singly housed and randomly assigned to receive water or water supplemented with 2% AKG (α-Ketoglutaric acid disodium salt, A610289; Sangon Biotech (Guangzhou) Co., Ltd). Body weight and food intake were monitored weekly for 6 or 11 weeks. At the end of the experiment, mice from both sexes were deeply anesthetized and euthanized. Gastrocnemius muscle and gWAT were isolated and weighed. An aliquot of gWAT was collected for HE staining and adipocyte size analysis. Additionally, serum and BAT were also collected in males. Serum was used to test the levels of NEFA, E, NE, T3, and T4. BAT was used to determine the mRNA and protein expression of UCP1, Dio2, and Cidea.

To investigate the metabolic effects of AKG in the DIO model, both male and female C57BL/6 mice at 12 weeks of age were singly housed and switched to HFD (D12492; Guangdong Medical Science Experiment Center). The mice were weighed and randomly assigned to receive water or water supplemented with 2% AKG. Body weight and food intake were continuously monitored weekly for 11 weeks. At the end of the experiment, body composition was determined using a nuclear magnetic resonance system (Body Composition Analyzer MiniQMR23-060H-I; Niumag Corporation, Shanghai, China). In females, mice were euthanized to collect and weigh BAT, gWAT, and iWAT. An aliquot of gWAT or iWAT was used for adipocyte size analysis. In males, core body temperature was measured using a RET-3 rectal probe (Kent Scientific, Torrington,

CT, USA). Then, mice were adapted into Promethion Metabolic Screening Systems (Sable Systems International, North Las Vegas, NV, USA). After adaptation for 5 days, $O_2$ consumption and RER were monitored for 3 days. $O_2$ consumption was normalized by body weight to represent energy expenditure. Subsequently, male mice were exposed to cold stress at 4°C for 6 h. BAT surface temperatures were recorded using a FLIR E60 thermal imaging camera (FLIR Systems, Wilsonville, OR, USA). Then, mice were euthanized to collect and weigh BAT, gWAT, and iWAT. An aliquot of gWAT or iWAT was used for adipocyte size analysis. Another aliquot of gWAT or iWAT was used to determine the mRNA expression of PPARγ, FASN, and ACC or protein expression of p-HSL and ATGL. BAT was used to determine the mRNA and protein expression of UCP1, Dio2, and Cidea. Serum was also collected to test the levels of AKG, NEFA, E, NE, T3, and T4.

### Acute effects of AKG on energy homeostasis

To examine the acute effects of AKG on BAT thermogenesis and WAT lipolysis, 12-week-old male C57BL/6 mice were weighed and randomly divided into two groups to receive i.p. injection of either saline or 10 mg/kg AKG, respectively. Immediately after i.p. injection, mice were exposed to cold stress at 4°C. After 6-h cold exposure, BAT surface temperatures were recorded as described before. One week after cold exposure, mice were concordantly i.p. injected with saline or 10 mg/kg AKG. Three hours after injection, the mice were euthanized to collect serum, iWAT, BAT, and adrenal gland. An aliquot of iWAT was used to determine the protein expression of ATGL and p-HSL. Serum was used to test E, NE, and NEFA levels. BAT was used to determine the mRNA and protein expression of UCP1, Dio2, and Cidea. The adrenal gland was used to determine the protein expression of PLCβ and p-Erk.

To examine the acute effects of AKG on physical activity and heart rate, another cohort of 12-week-old male C57BL/6 mice was used. Under anesthesia, a telemetric Mini-Mitter probe (G2 HR E-Mitter; Starr Life Science, Oakmont, PA, USA) was implanted into the abdominal cavity according to the manufacturer's instructions. Two weeks after recovery, mice were adapted into Promethion Metabolic Screening Systems. After adaptation for 5 days, mice were weighed and randomly divided into two groups to receive i.p. injection of saline or 10 mg/kg AKG at 7:00 am. The physical activity and heart rate were monitored in real time for 24 h by Promethion Metabolic Screening Systems. To examine if ADRB3 mediates the metabolic effects of AKG, 10-week-old male C57BL/6 mice were singly housed and adapted into Promethion Metabolic Screening Systems. After 5 days of adaptation, mice were weighed and randomly divided into four groups receiving i.p. injection of saline, 10 mg/kg AKG, 1 mg/kg SR59230A (S8688; Sigma-Aldrich), or AKG + SR59230A at 7:00 am. $O_2$ consumption and RER were continuously recorded for 24 h after injection as described above.

### AKG's effects in adrenalectomized mice

Eight-week-old male C57BL/6 mice were anesthetized with inhaled isoflurane. As described before (Makimura *et al*, 2000, 2003), these mice received bilateral adrenalectomy or sham surgery. To compensate for the lass of mineralocorticoids, drinking water was

supplemented with 0.9% NaCl. Two weeks after the surgery, when all mice recovered from body weight loss induced by surgery stress, mice from each surgery group were weighed and further divided into two groups receiving either water or water supplemented with 2% AKG. Body weight and food intake were continuously monitored weekly for 10 weeks. At the end of the experiment, cold-induced BAT thermogenesis was tested as described before. Body composition and weights of iWAT and gWAT were determined. An aliquot of iWAT was used to determine protein expression of p-HSL and ATGL. BAT was used to determine the mRNA and protein expression of UCP1, Dio2, and Cidea. Serum was also collected to test the levels of NEFA.

### AKG response in *ex vivo* cultured BAT

AKG's effects on oxygen consumption rate (OCR) were determined in *ex vivo* cultured BAT. Briefly, the dissolved oxygen rate was measured by Micro 4 Oxygen meter (PreSens Precision Sensing GmbH, Regensburg, Germany) at 0, 5, 15, 25, 35, 45, and 55 min after vehicle, 50 μM AKG, 100 μM AKG, or 10 μM NE treatment. The differences in dissolved oxygen rate between sessions represent OCR. In another separate trial, NEFA was measured in the supernatant medium from BAT after 30 min of *ex vivo* treatment with vehicle, 50 μM AKG, 100 μM AKG, or 10 μM NE.

### AKG response in chromaffin cells *in vitro* and adrenal glands *ex vivo*

To determine the effects of AKG on adrenal E release, adrenal gland was *ex vivo* cultured and treated with 0 or 100 μM AKG for 30 min. The supernatant medium was collected to test the levels of E and NE, while adrenal gland was collected for Western blot analysis of PLCβ protein expression. Similarly, chromaffin cells were treated with 0, 5, 50, 60, 80, or 100 μM AKG for 30 min. The supernatant medium was collected to test the levels of E and NE.

In another separate trial, the effects of AKG on intracellular calcium concentration were tested in chromaffin cells. Intracellular calcium was measured by calcium fluorometry following the manufacturer's instructions of Fluo-8-AM kit (AAT Bioquest, Sunnyvale, CA, USA). Briefly, chromaffin cells were washed twice with Hanks' balanced salt solution (HBSS, pH = 7.2–7.4) containing 8 g/l NaCl, 0.4 g/l KCl, 0.1 g/l $MgSO_4.7H_2O$, 0.1 g/l $MgCl_2.6H_2O$, 0.06 g/l $Na_2HPO_4.2H_2O$, 0.06 g/l $KH_2PO_4$, 1 g/l glucose, 0.14 g/l $CaCl_2$, and 0.35 g/l $NaHCO_3$ and incubated with 10 μM Fluo-8-AM at 37°C for 1 h. After incubation, cells were washed twice again with HBSS and incubated with vehicle, 100 μM AKG, 100 μM succinate, or 100 μM glutamate. Nikon Eclipse Ti-S microscope was used to observe fluorescence which was initiated by AKG, succinate, or glutamate that we added. Fluorometric data were acquired at excitation and emission wavelengths of 490 and 525 nm (490/525 nm) every 2 s over a 180-s period.

### Expression mapping of AKG sensors

Twelve-week-old male C57BL/6 mice were euthanized to collect adrenal gland, testis, hypothalamus, cortex, BAT, iWAT, gWAT, gastrocnemius muscle, soleus muscle, and liver. The mRNA expressions of OXGR1, PHD1, PHD2, PHD3, TET1, TET2, JMJD3, OCT4,

ASZ1, wdfc15a, and Dazl in these tissues were tested by absolute RT–PCR. An aliquot of adrenal gland was fixed in 4% paraformaldehyde and cut into 8-μm sections. Sections were then subjected to immunofluorescence staining of OXGR1.

### OXGR1 knockdown in chromaffin cells

The OXGR1 siRNA and negative control siRNA were purchased from GenePharma Co., Ltd. (Shanghai, China) and transfected into chromaffin cells using Lipofectamine reagents (Invitrogen, Carlsbad, CA, USA) according to the manufacturer's instructions. The sequences of siRNA targeting OXGR1 are 5′-CCGACGAGCAAAUCUCAUUTT-3′ (sense) and 5′-AAUGAGAUUUGCUCGUCGGTT-3′ (anti-sense). The sequences of negative control siRNA are 5′-UUCUCCGAACGUGUC ACGUTT-3′ (sense) and 5′-ACGUGACACGUUCGGAGAATT-3′ (anti-sense). An aliquot of transfected cells was collected to determine OXGR1 protein expression. Another aliquot of transfected cells was treated with 0 or 100 μM AKG for 30 min. The supernatant medium was collected to test the levels of E. The last aliquot of transfected cells was used for Fluo-8-AM assay to test AKG's effect on calcium flux as described above.

### OXGR1 knockout (KO) mouse model

OXGR1KO mouse model was generated by Shanghai Model Organisms Center, Inc. The guide RNAs targeting exon 4 of OXGR1 gene were designed using CRISPR/Cas9 strategy as shown in a specific scheme (Appendix Fig S1A). The Cas9 mRNA was *in vitro* transcribed using mMESSAGE mMACHINE T7 Ultra Kit (Ambion, TX, USA) according to the manufacturer's instructions. Two sgRNAs were designed to delete the OXGR1 protein-coding region using the online designer (http://crispr.mit.edu/). The target sequences of two sgRNAs were 5′-GTTTAACCTCTAACTTCCAC-3′ and 5′-TTAAA GGCTCGAAGGCTAAC-3′. The sgRNAs were *in vitro* transcribed using the MEGAshortscript Kit (Thermo Fisher, USA) and subsequently purified using MEGAclear™ Kit (Ambion, Life Technologies). The mixture of Cas9 mRNA and sgRNAs was co-injected into zygotes of C57BL/6 mouse by microinjection. F0 mice were genotyped by PCR, using the following primer pairs: Forward: 5′-TATACC AGCTGTTTTTCTTGTTGC-3′; Reverse: 5′-GATGCGTGGCTGTTTATG TCA-3′. The genotype of positive F0 was confirmed by sequencing. The positive F0 mice were chosen and crossed with C57BL/6 mice to produce F1 mice. The genotype of F1 mice was identified by PCR and confirmed by sequencing (Appendix Fig S1B). The mRNA expression of OXGR1 was also compared between WT and OXGR1KO mice. F1 mice with protein-coding region deletion in exon 4 were used to intercross to obtain the homozygous OXGR1KO mice.

To determine if OXGR1 mediates the metabolic effects of AKG, male OXGR1KO and WT control mice at 12 weeks of age were singly housed and switched to HFD. These mice were weighed and further divided into two groups receiving water or water supplemented with 2% AKG. Body weight was continuously monitored weekly for 13 weeks. At the end of the experiment, body composition was measured by QMR, while food intake, $O_2$ consumption, and RER were monitored by Promethion Metabolic Screening Systems as described before. Then, mice were euthanized to collect and weigh gWAT and iWAT. An aliquot of gWAT or iWAT was used for adipocyte size analysis

by HE staining. Another aliquot of both gWAT and iWAT was used to determine protein expression of p-HSL and ATGL. BAT was collected to determine the protein expression of UCP1. Serum was also collected to test the levels of E.

To determine if OXGR1 is required for the anti-obesity effects of exercise, exercise-induced metabolic beneficial effects were compared between WT and OXGR1KO mice. Specifically, male C57BL/6 WT control or OXGR1KO mice were switched to HFD at 8 weeks of age. After 12 weeks of HFD feeding, mice were further divided into two groups receiving non-exercise or resistance exercise. For resistance exercise group, mice first received 3 days of adaption and then 11 days of resistance training. Adaptation and resistance training were conducted as described before. Body weight and food intake of all groups were monitored every day. Body composition was measured at 0, 4, 8, and 14 days after exercise. At the end of the 14-day exercise, $O_2$ consumption and RER were monitored by Promethion Metabolic Screening Systems as described before. Then, mice were euthanized to collect and weigh gWAT and iWAT. An aliquot of iWAT was used to determine the mRNA expression of p-HSL and ATGL. BAT was collected to determine the mRNA expression of UCP1. Serum was also collected to test the levels of E. In a separate experiment, another cohort of male OXGR1KO mice (10 weeks old) received 40-min resistance exercise as described before. Serums were collected before and immediately after exercise.

### Adrenal-specific reexpression or overexpression of OXGR1

For the generation of OXGR1 overexpression HBAAV2/9-OXGR1 and control HBAAV2/9-GFP strains, OXGR1 coding region or GFP cassette was subcloned into the backbone of a pHBAAV-CMH-MCS-3flag-EF1-ZsGreen expression plasmid (Hanbio Biotechnology, Shanghai, China). Following DNA sequencing screening, the AAV plasmid was packaged into AAV serotype 2/9 virus by Hanbio Biotechnology. To specifically test if adrenal OXGR1 mediates the metabolic effects of AKG, OXGR1 adrenal-specific reexpression mouse model (GRPRE^AG) was generated by selectively delivering HBAAV2/9-OXGR1 into the adrenal gland of OXGR1KO mice. Briefly, 8-week-old male OXGR1KO mice were anesthetized with inhaled isoflurane. Bilateral incisions were made through the shaved skin of the abdominal wall just ventral to the kidney. Two microliters of HBAAV2/9-OXGR1 ($1.1 \times 10^{12}$ VG/ml) or control HBAAV2/9-GFP ($1.4 \times 10^{12}$ GC/ml) was bilaterally injected into the exposed adrenal glands. After 2 weeks of surgery recovery, mice were switched to HFD and further divided into two groups receiving water or water supplemented with 2% AKG for 12 weeks ($n = 8$ per group). The long-term effects of AKG on body weight gain, food intake, energy expenditure, RER, body composition, and lipolysis and thermogenesis of fat tissues were investigated as described in the OXGR1KO model. Adrenal glands were also collected to test the mRNA expression of OXGR1. Similarly, an adrenal-specific overexpression mouse model (OXGR1OE^AG) was generated by selectively delivering HBAAV2/9-OXGR1 into the adrenal gland of C57BL/6 WT mice. The metabolic effects of long-term AKG supplementation were compared between OXGR1OE^AG and control mice with an adrenal-specific injection of HBAAV2/9-GFP following the same experimental procedures as in the OXGR1RE studies.

## HE staining

HE staining was performed as described before (Zhu et al, 2017). Briefly, an aliquot of iWAT and gWAT was fixed with 10% formalin and embedded with paraffin. Then, fixed iWAT and gWAT were sectioned and stained with hematoxylin and eosin (HE). Pictures of stained adipose tissue were obtained in the same location with up to six fields of view. Adipocyte sizes of thirty adipocytes were analyzed per section. Data from 5 mice were averaged for each group.

## Western blot analysis

Western blot analysis was performed as described before (Zhu et al, 2017). Briefly, total protein lysates (20 μg) were immunoblotted with rabbit anti-OXGR1 antibody (1:2,000, LS-A1865; LifeSpan BioSciences, Inc., Seattle, WA, USA), rabbit anti-p-HSL (Ser563) (1:1,000, #4139; Cell Signaling), rabbit anti-HSL (1:500, sc-17194; Santa Cruz Biotechnology, Inc., Dallas, TX, USA), rabbit anti-ATGL (1:1,000, #2138; Cell Signaling), rabbit anti-UCP1 (1:1,000, #14670; Cell Signaling), rabbit anti-p-Erk (1:1,000, #4370; Cell Signaling), rabbit anti-Erk (1:1,000, #9102; Cell Signaling), rabbit anti-PLCβ (1:500; Santa Cruz), rabbit anti-p-AMPKα (Ser485) (1:1,000, #AP0116; ABclonal), rabbit anti-AMPKα (1:2,000, #A17290; ABclonal), rabbit anti-FoxO1 (1:1,000, #9454; Cell Signaling), rabbit anti-p-FoxO1 (Ser256) (1:1,000, #9461; Cell Signaling), rabbit anti-p-IKK (Ser176) (1:1,000, #2078; Cell Signaling), rabbit anti-IKK (1:1,000, #2682; Cell Signaling), rabbit anti-NF-κB (p65) (1:1,000, #8242; Cell Signaling), rabbit anti-p-IκB (Ser32) (1:1,000, #2859; Cell Signaling), rabbit anti-IκB (1:1,000, #4812; Cell Signaling), or rabbit anti-β-tubulin (1:50,000, AP0064; Bioworld Technology, Inc., St. Louis Park, MN, USA), followed by donkey anti-goat HRP-conjugated secondary antibody or goat anti-rabbit HRP-conjugated secondary antibody (1:50,000, bs-0294D or bs-0295G; Bioss, Woburn, MA, USA). The levels of tubulin served as the loading control.

## Relative quantitative PCR analysis

Real-time PCR assay was performed as described before (Cai et al, 2016). Briefly, total mRNA was extracted and digested with DNase I. The total mRNA (1 μg) was reverse-transcribed to cDNA using oligo (dT) 18 primer. SYBR Green relative quantitative real-time PCR was performed according to published protocols (Bookout & Mangelsdorf, 2003). Results were normalized by the expression of housekeeping gene β-actin.

## Absolute quantitative PCR analysis

The absolute quantitative RT–PCR assay was performed according to published protocols (Chini et al, 2007). Briefly, the cDNA samples of mouse tissues were first generated as described in relative quantitative RT–PCR analyses. The Ct value of each gene was obtained for further analysis. To generate a standard curve for each gene, the specific PCR amplification product was purified by electrophoresis and gel extraction using Agarose Gel Recovery kit (D2111-02; Magen BioSciences, Waltham, MA, USA). The DNA concentration of each product was measured by NanoDrop (2000c; Thermo Fisher Scientific). The absolute copy number of each sample was calculated according to the following formula: $C = A/B \times 6.02 \times 10^{14}$, where A is the concentration obtained by OD260 analysis (ng/μl), B is the molecular weight of the synthesized DNA (Daltons), and C is the copy number of the synthesized DNA (copies/μl). Subsequently, eightfold serial dilution was carried out on each purified PCR product 12 times. The dilutions of each product were used as the templates for SYBR Green quantitative real-time PCR to target gene using the above-mentioned primers amply. The Ct value of each dilution was obtained. The standard curve of each gene was plotted as a linear regression of the Ct values versus the log of the copy number. The final quantification data of each gene in different tissues were obtained by interpolating the Ct value into the standard curve.

## Immunofluorescence staining (IF)

Adrenal gland sections were incubated with the primary rabbit anti-OXGR1 antibody (1:1,000, LS-A1865; LSBio) at room temperature overnight, followed by goat anti-rabbit FITC-conjugated secondary antibody (1:1,000, bs-0295G; Bioss) for 1 h. Sections were mounted on slides and coverslipped with Mounting Medium with DAPI (H-1200; Vector Laboratories, Burlington, ON, Canada). Fluorescent images were obtained using a Nikon Eclipse Ti-S microscope (Nikon Instruments, Tokyo, Japan).

## NF-κB translocation

Adrenal gland chromaffin cells were cultured in six-well plates with an adhesive coverslip. About 50% coverage of coverslip, the cell was treated with 0 or 100 μM AKG for 3 h. OXGR1 was interfered with siOXGR1 as described above. IKK was inhibited by IKK inhibitor IKK16 (S2882; Selleck, USA) for 3 h. IKK signaling pathway was further tested by Western blot. Cell-climbing slices were rinsed three times in PBS, fixed in paraformaldehyde for 10 min, and washed in 0.4% Triton X-100 (T9284; Sigma) for 30 min. After 1 h of blockage in 3% goat serum at room temperature, the slices were incubated overnight in rabbit anti-NF-κB (p65) (1:1,000) at room temperature. The next day, the slices were rinsed three times in PBS and incubated in goat anti-rabbit FITC-conjugated secondary antibody (1:1,000, bs-0295G; Bioss). Then, the slices were coverslipped with Mounting Medium with DAPI (H-1200; Vector Laboratories, Burlington, ON, Canada). Fluorescent images were obtained using a Nikon Eclipse Ti-S microscope (Nikon Instruments, Tokyo, Japan).

## Immunohistochemical staining (IHC)

Adipose tissue IHC staining was performed as described before (Bal et al, 2017). Mouse adipose tissues (brown and white adipose tissue) were fixed with 10% formalin, embedded with paraffin, and sliced into 10-μm sections. The sections were mounted on slides and dehydrated with sequential alcohol gradient (0, 50, 70, 80, 90 and 100%). Antigen retrieval was performed by incubating with EDTA reagent at 90°C for 10 min. The sections were then incubated with 0.3% $H_2O_2$ for 30 min, followed by blocking solution buffer for 1 h at room temperature. Subsequently, the sections were incubated in rabbit anti-phospho-HSL (1:1,000) or rabbit anti-UCP1 (1:1,000) overnight, followed by Biotin-SP-AffiniPure Goat Anti-Rabbit IgG (111-065-003; Jackson ImmunoResearch, West Grove, PA, USA) for 1 h at room temperature. The sections were then visualized by

incubation with the ABC kit (PK-4000; Vector Laboratories, Burlingame, CA, USA) according to the manufacturer's instructions. Sections were then treated with diaminobenzidine (D12384; Sigma) for 5 min, followed by dehydration in a graded ethanol series from 50 to 100% and a final wash in xylene. Images were obtained using Upright microscopes, and Image-Pro Plus software was used to quantify grayscale. Up to six fields of view were captured from the same location within each adipose tissue.

### Hormone, metabolite, and enzyme activity assay

Serum levels of epinephrine (E), norepinephrine (NE), thyroxine (T4), triiodothyronine (T3), and non-esterified fatty acid (NEFA), as well as the enzyme activity of alpha-ketoglutarate dehydrogenase (OGDH), isocitrate dehydrogenase (ICDHm), and glutamate dehydrogenase (GDH) in different muscle tissues, were measured using commercial available kits according to the manufacturer's instructions (Nanjing Jiancheng Bioengineering Institute, Nanjing, China).

### Muscle contraction experiment *in vivo*

The method was performed as described previously (Park *et al*, 2012; Ato *et al*, 2016). For *in vivo* gastrocnemius electrical stimulation, 10-week-old male C57BL/6 mice were anesthetized by isoflurane, and the gastrocnemius muscles were surgically exposed. The soleus muscle was removed. Subsequently, the mice were positioned with their right foot on a footplate. The gastrocnemius muscle was connected to an electrical stimulator with an isolator. The insulation pad was adjusted to 37°C to cover the muscle during the whole experimental period of time. Physiological solution (2.5 mM $Ca^{2+}$ Tyrode solution: 5 mM KCl, 140 mM NaCl, 10 mM HEPES, 2 mM $MgCl_2$, 2.5 mM $CaCl_2$, and 10 mM glucose) was used to infiltrate muscle. Supramaximal electricity with a pulse width of 1 ms was delivered to muscles by a pair of platinum electrodes placed in parallel. The stimulation parameters were as follows: The wave width was 1 ms, the delay was 100 ms, the single continuous stimulus was used, the stimulation frequency was 50 Hz, and the stimulation intensity gradually was increased from zero under the condition of continuous single stimulation. Electrical stimulation was performed in unilateral gastrocnemius for 40 min (10 times, each time for 4 min, resting for 2 min between stimulations). The effect of temperature on muscle contraction characteristics can be effectively controlled during the experiment to prevent muscle inactivation. The contractile performance was assessed by measuring half-relaxation time (the time required for the force to decrease 50% from the peak value at the end of stimulation). The *in vivo* contractility experiment was set up using the BL-420F biological signal acquisition and analysis system (Chengdu Taimeng Software Co., Ltd. China). After stimulation, the mice were sacrificed, and plasma was collected to analyze the enrichment of AKG.

### Primary brown adipocyte preparation and differentiation

The interscapular brown adipose stromal vascular fraction was obtained from 6-week-old male C57BL/6 mice as described previously (Mills *et al*, 2018). In brief, interscapular brown adipose tissue was dissected and washed in PBS. Then, it was minced and digested for 45 min at 37°C in PBS (1.5 mg/ml collagenase B, 123 mM NaCl, 5 mM KCl, 1.3 mM $CaCl_2$, 5 mM glucose, 100 mM HEPES, and 4%

essentially fatty-acid-free BSA). The obtained tissue suspension was filtered through a 40-μm cell strainer and then centrifuged at 600 *g* for 5 min to pellet the SVF. After centrifugation, the upper layer of liquid (grease layer) was gently blotted dry; then, the medium layer was blotted dry, so as not to affect the underlying precipitate (red cells on the pellet), and then, 10 ml of SVF (10% FBS) medium was added. The 10-ml pipette was blown up and down 5–10 times, and the precipitate was blown off, then passed through a 40-μm filter sieve, mixed with three volumes of SVF (10% FBS) medium, and centrifuged at 600 g for 5 min. The culture solution was poured off, and the precipitate was retained. Cells were resuspended in 8–10 ml SVF (10% FBS) medium. The dish was evenly blown and inoculated into a 10-cm² petri dish to inoculate the cells uniformly. The cell pellet was resuspended in brown adipocyte culture medium and plated evenly. Cells were maintained at 37°C in 10% $CO_2$. (The above operations were all started in the afternoon or evening within 90 min.) The next morning (about 12 h), the culture medium of the culture dish was removed, and the cells were washed 4 times in warm PBS (4 ml) (fast cross-shake for 5–10 s). The amount of impurities was determined whether to continue washing (normal cells are adherent spindles, and angular, transparent, and floating cells are not), and after washing, 10 ml of SVF (10% FBS) medium was added. The following morning, brown pre-adipocytes were induced to differentiate with an adipogenic cocktail (0.5 mM isobutylmethylxanthine, 5 mM dexamethasone, 320 nM insulin, 1 nM triiodothyronine, and 0.125 mM indomethacin) in the adipocyte culture medium. Two days after induction, cells were re-fed every 48 h with adipocyte culture medium containing 1 μM rosiglitazone, 1 nM T3, and 0.5 μg/ml insulin. Cells were fully differentiated by day 7 after induction.

### OCR

Cellular OCR was determined using an Agilent Seahorse XFp analyzer (S7802A; Agilent Technologies). The culture of C2C12 and HepG2 cells was described previously (Cai *et al*, 2018; Xu *et al*, 2018). C2C12 (China Infrastructure of Cell Line Researcher, China) and primary brown adipocytes were plated and differentiated in Seahorse XFp cell culture miniplates (103022100; Agilent Technologies). Adrenal gland medulla cells and HepG2 cells were plated and differentiated in Seahorse XFp cell culture miniplates. The assay medium was prepared by supplementing Agilent Seahorse XF Base Medium (102353-100, 103193-100, 103334-100; Agilent Technologies). Agilent Seahorse recommends 1 mM pyruvate (S8636; Sigma), 2 mM glutamine (G8540; Sigma), and 10 mM glucose (G8769; Sigma) as a starting point. Before analysis, the adipocyte culture medium was changed to respiration medium consisting of DMEM lacking $NaHCO_3$ (Sigma), NaCl (1.85 g/l), phenol red (3 mg/l), 2% fatty-acid-free BSA, and sodium pyruvate (1 mM), and the pH was adjusted to 7.4. Basal respiration was determined to be the OCR in the presence of substrate (1 mM sodium pyruvate) alone. The assay medium was warmed to 37°C. The pH was adjusted to 7.4 with 0.1 N NaOH. Oligomycin inhibits ATP synthase (complex V), and the decrease in OCR relates to the mitochondrial respiration associated with cellular ATP production after injection of oligomycin (1 μM). Spare respiratory capacity, defined as the difference between maximal and basal respiration, can be calculated by the carbonyl cyanide-4-(trifluoromethoxy)phenylhydrazone (FCCP) (2 μM)-stimulated OCR. The combination of complex I

inhibitor rotenone (0.5 µM) and complex III inhibitor antimycin A (0.5 µM) can shut down mitochondrial respiration and enable the calculation of nonmitochondrial respiration driven by processes outside the mitochondria. The Agilent Seahorse XF Cell Mito Stress Test Report Generator automatically calculates the Agilent Seahorse XFp Cell Mito Stress Test parameters from Wave data.

## Bomb calorimetry of feces

Calorimetry was conducted using a calorimeter (IKA C200, Germany) as described previously (Mills *et al*, 2018). Male C57BL/6 mice were fed HFD, and fecal specimens were collected over a 48-h period. Collected fecal samples were baked at 60°C for 24 h to remove water content. Fecal samples were combusted, and the energy content of the fecal matter was measured as the heat of combustion (kJ/g).

## Blood pressure test

Mouse blood pressure was tested by a non-invasive blood pressure measurement system with the biological signal acquisition and analysis system (BP300A; Chengdu Taimeng Software Co., Ltd. China). The mice were restrained on a fixed frame and placed on a 37°C thermostat pad. After 10 min of acclimation, the blood pressure sensor was placed on the root of the mouse tail. Then, the systolic blood pressure and diastolic blood pressure of the mice were monitored and recorded in real time.

## 16S rDNA sequencing

Fresh feces of mice were collected at 9:00 a.m. and frozen rapidly at −80°C. 16S sequencing was carried out at Beijing Novogene Co., Ltd. The experimental method was described in the previous literature (Chen *et al*, 2017). Total genome DNA from samples was extracted using the CTAB/SDS method, and the concentration and purity were monitored by 1% agarose gels. The processed DNA was diluted to 1 ng/µl by sterile water. Amplicon Generation: 16S rRNA/18S rRNA/ITS genes of distinct regions (e.g., 16S V3-V4, 18S V4, ITS1/ITS2, Arc V4) were amplified by specific primers (e.g., 16S V3-V4: 341F-806R, 18S V4: 528F-706R) with the barcode. The PCRs were carried out in 30 µl reactions consisting of 15 µl of Phusion High-Fidelity PCR Master Mix (New England Biolabs), 0.2 µM of forward and reverse primers, and 10 ng of template DNA sample. The procedure of thermal cycling was as follows: 98°C for 1 min, followed by 30 cycles of denaturation at 98°C for 10 s, annealing at 50°C for 30 s, and elongation at 72°C for 30 s, with a final extension at 72°C for 5 min. Then, PCR products were mixed with same volume of 1× loading buffer (containing SYBR Green) and detected on 2% agarose gel for electrophoresis. PCR products were mixed in equidensity ratios, and the PCR product mix was purified with Gene-JET™ Gel Extraction Kit (Thermo Scientific). Library preparation and sequencing were carried out by the Ion Plus Fragment Library Kit 48 rxns (Thermo Scientific), and the library quality was assessed on the Qubit 2.0 Fluorometer (Thermo Scientific). At last, the library was sequenced on an Ion S5™ XL platform, and 400-bp/600-bp single-end reads were generated. Sequence analyses were performed using Uparse software (Uparse v7.0.1001, http://drive5.com/uparse/), and 97% similarity was considered to the same OTUs.

Representative sequence for each OTU was screened for further annotation. The Silva database (https://www.arb-silva.de/) was used based on the Mothur algorithm to annotate taxonomic information. Beta diversity analysis was used to evaluate differences of samples in species complexity. Beta diversity on both weighted and unweighted unifrac were calculated by QIIME software (Version 1.7.0). ANOSIM and ADONIS use the ANOSIM and ADONIS functions of the R vegan package, respectively.

## Transcriptomics

Samples from adrenal chromaffin cells were used for transcriptomic signature analysis. The untargeted transcriptomics profiling was performed on the Illumina platform (Novogene, Beijing, China) by Novogene Co., Ltd (Beijing, China). The sample preparation procedures were according to the previously published methods with minor modifications (Parkhomchuk *et al*, 2009). RNA was extracted by a RNA extraction kit (Magen, China). The RNA integrity was assessed by the RNA Nano 6000 Assay Kit of the Bioanalyzer 2100 system (Agilent Technologies, CA, USA). Total RNA (3 µg per sample) was used for the RNA sample preparations. Sequencing libraries were generated using the NEBNext® Ultra™ RNA Library Prep Kit for Illumina® (NEB, USA) following the manufacturer's recommendations, and index codes were added to attribute sequences to each sample.

Briefly, mRNA was fist purified by poly-T oligo-attached magnetic beads. Fragmentation was carried out using divalent cations under elevated temperature in the NEBNext First Strand Synthesis Reaction Buffer (5X). Next, first-strand cDNA was synthesized using random hexamer primer and M-MuLV Reverse Transcriptase (RNase H), and the second-strand cDNA synthesis was subsequently performed using DNA polymerase I and RNase H. The remaining overhangs were converted into blunt ends via exonuclease/polymerase activities. After adenylation of 3' ends of DNA fragments, NEBNext Adaptor with hairpin loop structure was ligated to prepare for hybridization. The library fragments were purified with the AMPure XP system (Beckman Coulter, Beverly, USA) to obtain cDNA fragments of preferentially 250–300 bp in length. Finally, 3 µl USER Enzyme (NEB, USA) was used with size-selected, adaptor-ligated cDNA at 37°C for 15 min, then at 95°C for 5 min, before PCR. PCR was performed with Phusion High-Fidelity DNA polymerase, Universal PCR primers, and Index (X) Primer. At last, the Agilent Bioanalyzer 2100 system was used to purify the PCR products and evaluate the library quality. cBot Cluster Generation System (TruSeq PE Cluster Kit v3-cBot-HS; Illumia) and Illumina HiSeq platform were used for the clustering of the index-coded samples and the library preparations sequenced after cluster generation, and 125-bp/150-bp paired-end reads were generated. Gene networks representing key genes were identified by using Ingenuity Pathways Analysis v7.6 (IPA; Ingenuity Systems).

## Endurance exercise training

The method was performed as described previously (Luo *et al*, 2012). In brief, the mice engaged in strenuous exercise on a treadmill (47300 TREADMILL; Ugo Basile, Italy) and standard running test to reduce obesity. Eight-week-old male C57BL/6 mice were switched to HFD for 12 weeks. At 20 weeks of age, the mice were acclimated to the treadmill with a 5-min run at 10 m/min once daily for 2 days. The endurance test regimen was 10 m/min for the first

20 min, followed by 1 m/min increments at about 20-min intervals. The mice were considered exhausted when they were unable to avoid repeated electrical shocks. Exercise endurance capacity was equated with the total running distance achieved before exhaustion. The endurance exercise was conducted for 14 days. Exercise time, distance, body weight, and food intake were record every day.

### UPLC-Orbitrap-MS/MS analysis for metabolites

The methods were performed as described previously (Shi *et al*, 2016; Hui *et al*, 2017; Xin *et al*, 2018; Go *et al*, 2019). After thawing, the samples were fully homogenized by vortexing for 2 min. Tissue samples were fully ground and homogenized. One hundred microliters of serum or tissue samples was transferred in a 1.5-ml EP microtube, and 500 μl of methanol (mass spectrometry grade) was added to each sample to remove protein. All samples were vortexed for 2 min and centrifuged at 20,000 *g* and 4°C for 15 min. Five hundred microliters of the supernatant was taken and dried under nitrogen at normal temperature. After drying, 200 μl of methanol was added to each sample to reconstitute. All samples were vortexed for 2 min and then centrifuged at 20,000 *g* and 4°C for 15 min. The supernatant was transferred to a sample vial and stored at −80°C for testing. Serum and tissue extracts were analyzed using LC–MS/MS analysis (Uplc1290-6470A QQQ liquid chromatography–mass spectrometry instrument; Agilent Technologies).

For α-ketoglutarate (AKG), succinate (SUC), fumaric acid (FUMA), oxaloacetic acid (OAA), alpha-ketoisovaleric acid (α-keval), 2-hydroxy-3-methylbutyric acid (2H3MA), α-ketoleucine (α-kehex), pyruvic acid (Pyr), and malate measurements, the following parameters were used: separation column: C18 column; and column temperature: 40°C. Mobile phase: B: 100% acetonitrile; and A: hydrogen peroxide + 0.2% formic acid. Elution gradient: constant elution 99% A and 1% B. Flow velocity: 0.3 ml/min. Sample volume: 5 μl. Mass spectrometry conditions and parameters: detector: MS QQQ mass spectrometer. Ion source: ESI source; spray voltage: 4000 (+), 3500 (−); atomization temperature: 300 degrees; atomizing gas (sheath gas) pressure: 10arb; scanning mode: negative ion multi-reaction detection (MRM). Compound: MAL parent ion: 133.1; and daughter ion: 115.1. SUC mother ion: 117.1; and daughter ion: 73.1. AKG mother ion: 145.0; and daughter ion: 57.1.

For xanthine and hypoxanthine measurement, the following parameters were used: separation column: C18 column; and column temperature: 40°C. Mobile phase: B: 100% acetonitrile; and A: hydrogen peroxide + 0.2% formic acid. Elution gradient: constant elution 95% A and 5% B. Flow velocity: 0.4 ml/min. Sample volume: 5 μl. Mass spectrometry conditions and parameters: detector: MS QQQ mass spectrometer. Ion source: ESI source; spray voltage: 4,000 (+) and 3,500 (−); atomization temperature: 350 degrees; atomizing gas (sheath gas) pressure: 10arb; scanning mode: positive ion multi-reaction detection (MRM). Compound: xanthine parent ion: 153.1; daughter ion: 110.1. Hypoxanthine mother ion: 137.1; and daughter ion: 110.1.

### Plasma lactate measurement

Serum levels of lactate were measured using commercial available kits according to the manufacturer's instructions (Nanjing Jiancheng Bioengineering Institute, Nanjing, China).

### Nuclear protein extraction

For the nuclear or cytoplasmic protein extraction, proteins were isolated according to the procedure of the nuclear extraction kit (Solarbio, SN0020).

### Statistics

Statistical analyses were performed using GraphPad Prism 7.0 software (Chicago, IL, USA). Methods of statistical analyses were chosen based on the design of each experiment and are indicated in the figure legends. The data are presented as mean ± SEM. $P \leq 0.05$ was considered to be statistically significant.

## Data availability

The RNA-seq datasets produced in this study are available via open science framework: https://www.ncbi.nlm.nih.gov/bioproject/ PRJNA602040 accession number: PRJNA602040.

**Expanded View** for this article is available online.

## Acknowledgements

This work was supported by grants from the National Key Point Research and Invention Program (2018YFD0500403 to G. S.), National Natural Science Foundation of China (31790411 to Q. J. and 31572480 to G.S.), National Key Point Research and Invention Program (2016YFD0501205 to G.S.), Research and Development Projects in Key Areas of Guangdong Province (2019B020218001 to G.S.), Innovation Team Project in Universities of Guangdong Province (2017KCXTD002 to G. S.), and National Institute of Diabetes and Digestive and Kidney Diseases from National Institutes of Health (R00DK107008 to P. X., K01DK111771 to Y. J.). We wish to thank Shanghai Model Organisms Center for generating OXGR1KO mouse line, Metabo-Profile Biotechnology for metabolomics analysis, and Beijing Novogene Co., Ltd for transcriptomics analysis and 16S rDNA sequencing.

## Author contributions

YY, PX, and QJ were the main contributors to the conduct of the study, data collection and analysis, data interpretation, and manuscript writing; XC, TW, WP, JS, CZhu, CZha, DY, ZH, JY, YZ, MD, CY, FL, LL, GX, and FZ contributed to the conduct of the study; SW, LW, XZ, LI, SS, YJ, JW, JS, QW, PG, QX, and YZ contributed to the manuscript writing and data interpretation; and GS contributed to the study design, data interpretation, and manuscript writing.

### Conflict of interest

The authors declare that they have no conflict of interest.

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
