## [Review Process File · The EMBO Journal]

Exercise-induced α -ketoglutaric acid stimulates muscle hypertrophy and fat loss through OXGR1-dependent adrenal activation

Yexian Yuan, Pingwen Xu, Qingyan Jiang, Xingcai Cai, Tao Wang, Wentong Peng, Jiajie Sun, Canjun Zhu, Cha Zhang, Dong Yue, Zhihui He, Jinping Yang, Yuxian Zeng, Man Du, Fenglin Zhang, Lucas Ibrahim, Sarah Schaul, Yuwei Jiang, Jiqiu Wang, Jia Sun, Qiaoping Wang, Liming Liu, Songbo Wang, Lina Wang, Xiaotong Zhu, Ping Gao, Qianyun Xi, Cong Yin, Fan Li, Guli Xu, Yongliang Zhang, Gang Shu

Review timeline:

Submission date:	5th Oct 2019
Editorial Decision:	4th Nov 2019
Revision received:	23rd Dec 2019
Editorial Decision:	22nd Jan 2020
Revision received:	25th Jan 2020
Accepted:	28th Jan 2020

Editor: Daniel Klimmeck

Transaction Report:

1st Editorial Decision

4th Nov 2019

Thank you for the submission of your manuscript (EMBOJ-2019-103304) to The EMBO Journal. Your manuscript has been sent to three reviewers, however, please note that one referee was not able to send us his/her comments as to unexpected additional obligations. We did receive reports from the other two reviewers, which I enclose below.

As you will see, the reviewers acknowledge the interest and novelty of your results, although they also express a number of issues that will have to be conclusively addressed before they can be supportive of publication of your manuscript in The EMBO Journal. Given the referees' recommendations, I would like to invite you to submit a revised version of the manuscript, addressing their criticism.

REFeree REPORTS:

Referee #1:

The paper describes the role of AKG as a secreted myometabolite on systemic metabolism. The authors show that AKG via the adrenal leads to increased circulating E levels and induces brown fat function and lipolysis. In addition, they identify Oxgr1 as the receptor which mediates this effect. The paper contains a fast amount of in-depth data which is presented in a very systematic manner. I have listed a few minor points which should be addressed in a revised version.

1. Fig. 1 Please present lean and fat mass independently.
2. The human data looks very interesting, please use a multivariate analysis to assess the contribution of AKG to the different risk factors

3. Please show 3E and G using a scale starting from 0
4. 3N-O and all other quantifications of BAT function, please include a Wblot of Ucp1
5. Change figure 3 to fit with the text, in its current form it is cited out of order
6. Increased circulating E levels should lead to enhanced browning in the sc AT. This should be addressed and discussed especially since it can be expected that brite cells are formed by interconversion.

Referee #3:

The manuscript by the Shu group proposes that α -ketoglutarate (AKG) is a metabolite whose levels increase in the bloodstream after resistance exercise and exert a beneficial effect on lean mass after high fat diet (HFD). They show that the effects of AKG are due to the stimulation of OXGR1 receptors and epinephrine release from the adrenal gland. The study contains an impressive list of experimental approaches spanning from the physiology of muscle exercise, the metabolomics analysis, the whole body energy homeostasis and the inter-organ cross-talk of metabolites and hormones as validated by novel OXGR1 mouse mutants. The study is therefore complete as it is, with 8 multi-panel main figures and 8 extended views.

Minor points:

- 1) The authors report that AKG derivatives also change after resistance training (Fig. 1F). It would be interesting to see how these metabolites vary after AKG administration.
- 2) It would be clearer to have a direct comparison of AKG levels in WT and KO mice (Fig. 7H)
- 3) In the discussion, the authors could mention that the data on the NF- κ B pathway (Fig. 8) awaits in vivo validation in mice.

1st Revision - authors' response

23rd Dec 2019

The authors greatly appreciate the critiques and constructive suggestions that helped to improve this manuscript submitted to EMBO J. In the initial review, this manuscript was praised for “a fast amount of in-depth data”, “presented in a very systematic manner”, and “an impressive list of experimental approaches”. All concerns have been addressed as detailed below point-to-point. Major revisions have been labelled with red lines by the left side margin in the revised manuscript.

Referee #1:

The paper describes the role of AKG as a secreted myometabolite on systemic metabolism. The authors show that AKG via the adrenal leads to increased circulating E levels and induces brown fat function and lipolysis. In addition, they identify Oxgr1 as the receptor which mediates this effect. The paper contains a fast amount of in-depth data which is presented in a very systematic manner. I have listed a few minor points which should be addressed in a revised version.

1. Fig. 1 Please present lean and fat mass independently.

This point is well taken. Fig. 1B has been re-organized. Lean and fat mass has been independently presented.

2. The human data looks very interesting, please use a multivariate analysis to assess the contribution of AKG to the different risk factors

We highly appreciate this point. The contribution of AKG to the different metabolic risk factors, including BMI, HCF, WCF, Fat, Weight, NCF, VF, Systolic, Diastolic, Height and Age has been assessed by a multivariate analysis. The results have been summarized in Fig. EV1H.

3. Please show 3E and G using a scale starting from 0

We appreciate the point. The scale bar has revised to start from 0 in Fig. 3C and E (re-organized from Fig. 3E and G).

4. 3N-O and all other quantifications of BAT function, please include a Wblot of Ucp1

This is an excellent point. Western quantification of UCP1 protein levels have been added in Fig. 3G. Other mouse models involving quantifications of BAT function have been listed as follows:

- (1) Acute AKG effects on BAT functions in WT mice: Fig. 4B-C (BAT temperature); Fig. EV4D (Wblot of UCP1).
- (2) Long-term AKG water supplementation effects on BAT functions in adrenalectomized WT mice: Fig. 4O-P (BAT temperature); Fig. 4Q and 4T-U (RT-PCR and Wblot of UCP1).
- (3) Long-term AKG water supplementation effects on BAT functions in OXGR1KO mice: Fig. EV7B (Wblot of UCP1).
- (4) Long-term AKG water supplementation effects on BAT functions in OXGR1RE_{AG} mice: Fig. EV7I (Wblot of UCP1).
- (5) Long-term AKG water supplementation effects on BAT functions in OXGR1OE_{AG} mice: Fig. EV8J (Wblot of UCP1).
- (6) Resistance exercise effects on BAT functions in OXGR1KO mice: Fig. 7J-K (RT-PCR and Wblot of UCP1).

5. Change figure 3 to fit with the text, in its current form it is cited out of order

We highly appreciate this point. We have revised Figure 3 to fit with the described text.

6. Increased circulating E levels should lead to enhanced briteening in the sc AT. This should be addressed and discussed especially since it can be expected that brite cells are formed by interconversion.

This is an excellent point. The effects of AKG water supplementation on cold-induced briteening in the sc AT has been tested. The results have been summarized in Fig. EV3E-I. We found that AKG increased brite adipocyte formation at room temperature and enhanced cold-formed brite adipocytes that expressed UCP1 in the iWAT. The following discussion has been added.

“While activation of the BAT thermogenesis has clinically significant effects, BAT is found in negligible volumes in adult humans (Yoneshiro, Aita et al., 2013). Indeed, cold exposure does not induce measurable metabolic response of small BAT depots in humans (Chondronikola, Volpi et al., 2014). Instead, data from rodents and humans (Betz & Enerback, 2015, Wu, Bostrom et al., 2012) suggest BAT-like white adipose tissue (beige or brite) as an intriguing and potentially more physiologically significant target for human metabolic syndrome. Indeed, human pre-adipocytes can be differentiated into brite adipocytes, which resemble classical brown adipocytes, and respond to cold adaptation or other stimuli (Betz & Enerback, 2015, Giralt & Villarroya, 2013, Rosenwald & Wolfrum, 2014). Interestingly, increased circulating E levels have been shown to enhance briteening in the subcutaneous white adipose tissue in both mice and humans (Cannon & Nedergaard, 2004, Kajimura, Spiegelman et al., 2015, Sidossis, Porter et al., 2015). Consistently, we found that water supplementation of AKG

significantly increased brite adipocytes that expressed UCP1 and other brite and thermogenic genes in the iWAT. These data suggest that brite, in rodents, may have therapeutic relevance, which can be enhanced by AKG supplementation. ”

Referee #3:

The manuscript by the Shu group proposes that α -ketoglutarate (AKG) is a metabolite whose levels increase in the bloodstream after resistance exercise and exert a beneficial effect on lean mass after high fat diet (HFD). They show that the effects of AKG are due to the stimulation of OXGR1 receptors and epinephrine release from the adrenal gland. The study contains an impressive list of experimental approaches spanning from the physiology of muscle exercise, the metabolomics analysis, the whole body energy homeostasis and the inter-organ cross-talk of metabolites and hormones as validated by novel OXGR1 mouse mutants. The study is therefore complete as it is, with 8 multi-panel main figures and 8 extended views.

Minor points:

1) The authors report that AKG derivatives also change after resistance training (Fig. 1F). It would be interesting to see how these metabolites vary after AKG administration.

We appreciate this constructive suggestion. We examined the levels of AKG derivatives (SUA: Succinic acid; FUMA: Fumaric acid; OAA: oxaloacetic acid; α -keval: Alpha-ketoisovaleric acid; 2H3MA: 2-Hydroxy-3-methylbutyric acid; α -kehex: α -ketoleucine; Pyr: Pyruvic acid) in the circulating blood 3 hrs after acute i.p. injection of AKG (10 mg/kg). We found that AKG increased the level of α -kehex and decreased the level of FUMA. The results have been summarized in Fig EV4M-S.

2) It would be clearer to have a direct comparison of AKG levels in WT and KO mice (Fig. 7H)

This is an excellent point. We tested serum AKG levels of male WT and OXGR1KO mice before or after 40 min resistance exercise. The results have been summarized in Fig. 7H. We found that the stimulatory effect of resistance exercise on serum AKG was attenuated by OXGR1KO. We speculated that this attenuation is attributed to the reduction of total muscle mass induced by OXGR1KO when mice were fed on HFD.

3) In the discussion, the authors could mention that the data on the NF- κ B pathway (Fig. 8) awaits in vivo validation in mice.

Revise : We highly appreciate this point. The following discussion has been added.

“As the first step to explore the intracellular mechanism of AKG-induced E release, we performed transcriptome analysis on adrenal chromaffin cells before or after AKG treatment. Interestingly, inflammatory signaling pathways were identified as the transcriptome signature induced by AKG treatment. We further provided in vitro evidence to support a model in which AKG acts through OXGR1 to induce NF- κ B signal transduction cascade, and by doing so, it stimulates E release from chromaffin cells. Consistent with this model, pro-inflammatory cytokines, tumor necrosis factor α , and interleukin 1 β / α have been shown to modulate catecholamine secretion and long-term gene regulation in chromaffin cells of adrenal medulla (Jenkins, Sreenivasan et al., 2016, Rosmaninho-Salgado, Araujo et al., 2009, Tamura, Nemoto et al., 2014). The activation of a cytokine-specific intracellular signaling pathway and following E2 release in adrenal medulla may be one part of a systemic stress response induced by exercise. Chromaffin cells may act as an integrative center responding to both immune and endocrine stimuli induced by exercise. Our lab is currently validating this model and exploring if p65/NF- κ B inflammatory pathway is required for the stimulatory effects of AKG on E release of adrenal chromaffin cell in vivo.”

2nd Editorial Decision

22nd Jan 2020

Thank you for submitting your revised manuscript for consideration by The EMBO Journal. Your revised study was sent back to the two referees for re-evaluation, and we have received comments from both of them, which I enclose below. As you will see the referees find that their concerns have been sufficiently addressed and they are now broadly in favour of publication.

Thus, we are pleased to inform you that your manuscript has been accepted in principle for publication in The EMBO Journal, pending some minor issues related to formatting and data representation points listed below, which need to be adjusted at re-submission.

REFeree REPORTS:

Referee #1:

The authors addressed all my comments very convincingly, the paper in my opinion should be accepted for publication.

Referee #3:

The authors addressed the previously raised issues

2nd Revision - authors' response

25th Jan 2020

The authors performed all requested editorial changes.

3rd Editorial Decision

28th Jan 2020

Thank you for submitting the revised version of your manuscript. I have now evaluated your amended manuscript and concluded that the remaining minor concerns have been sufficiently addressed.

Thus, I am pleased to inform you that your manuscript has been accepted for publication in the EMBO Journal.

Corresponding Author Name: Gang Shu

Journal Submitted to: The EMBO Journal

Manuscript Number: EMBOJ-2019-103304R